# Characterization of liquid cloud profiles using global collocated active radar and passive polarimetric cloud measurements

Yutong Wang<sup>1,2</sup>, Huazhe Shang<sup>1</sup>, Chenqian Tang<sup>3</sup>, Jian Xu<sup>4</sup>, Tianyang Ji<sup>5</sup>, Wenwu Wang<sup>1,2</sup>, Lesi Wei<sup>1</sup>, Yonghui Lei<sup>1</sup>, Jiancheng Shi<sup>4</sup>, Husi Letu<sup>1</sup>

- State Key Laboratory of Remote Sensing and Digital Earth, Aerospace Information Research Institute, Chinese Academy of Sciences, Beijing 100101, China
  - <sup>2</sup>University of Chinese Academy of Sciences, Beijing 100049, China
  - <sup>3</sup>Department of Geosciences, University of Oslo, Oslo 0371, Norway
  - <sup>4</sup>National Space Science Center, Chinese Academy of Sciences, Beijing 100190, China
- 10 <sup>5</sup>College of Computer Science, Inner Mongolia University, Hohhot 010021, China

Correspondence to: Huazhe Shang (shanghz@radi.ac.cn), Husi Letu (husiletuw@hotmail.com)

Abstract. Stratiform liquid cloud profiles are key to deciphering cloud life cycles, microphysical processes, and climate change impacts. Nevertheless, remote sensing of cloud vertical structure remains largely unresolved. CloudSat active measurements provide cloud microphysical profile products but are restricted to narrow orbital tracks. Multiangle passive imagers, such as Polarization and Directionality of Earth's Reflectance (POLDER), are capable of generating a variety of cloud properties with broad area coverage; however, they lack key prior knowledge and effective methods for obtaining cloud vertical information. Focusing on single-layer cloud profile retrieval, we first reveal the structural characteristics of stratiform cloud effective radius (CER) profiles based on global CloudSat data and find that the dominant structures include triangle-shaped and monotonically decreasing profiles, which account for approximately 88.5% of global liquid CER profiles. Furthermore, we propose a novel approach to estimate the structural characteristics of triangle-shaped profiles from POLDER observations like the properties of the profile turning point (TP). This approach integrates vertical structure morphology recognition with a combination of fitting methods and machine learning models. The cloud profiles are then accurately reconstructed using physical parameterization models. Our retrieval results exhibit good consistency with active observations, with an RMSE of 1.1 µm for TP CER and 0.1 for the normalized cloud optical thickness at the TP. This research advances the parameterization of liquid cloud profiles and enables profile structural characteristic retrieval based on a multiangle passive imager. Our findings provide valuable insights into improving the understanding and modelling of cloud processes in weather and climate systems.

# 1 Introduction

Low stratiform liquid clouds, which cover nearly 30% of the Earth's surface (Warren et al., 1986; Warren et al., 1988; Warren et al., 2007; Wood, 2012; Wood, 2015), play a crucial role in the climate system due to their extensive coverage and significant radiative effects, including reflecting shortwave solar radiation and absorbing longwave radiation from the Earth

(Slingo, 1990; Chen et al., 2000; Greenwald et al., 1995). These clouds are a key component of climate and must be accurately represented in general circulation models (GCMs) (Dong and Minnis, 2023; Turner et al., 2007). Compared with ice clouds, which have complex vertical structures and high uncertainty, stratiform liquid clouds are relatively homogeneous in the horizontal direction and have a certain thickness (Zhang et al., 2010; Mace et al., 2009). Therefore, these clouds are relatively ideal for studying cloud properties and growth processes using satellite observations and models.

35

The cloud microphysical profile is an indispensable parameter for describing cloud vertical structures, such as the profile of cloud effective radius (CER) and liquid water content (LWC) (Chen et al., 2008). Studies have demonstrated that cloud vertical structures are closely linked to the cloud life cycle and precipitation characteristics, atmospheric circulation, cloud–precipitation microphysical processes, and conditions for artificial rainfall (Nakajima et al., 2010a, b; Zhao et al., 2024; Breon and Doutriaux-Boucher, 2005; Sinclair et al., 2021). Accurately and comprehensively detecting and quantifying the vertical structural characteristics and geographical distribution of clouds is highly important for reducing uncertainties in the impact of clouds on climate change and exploring the role of clouds and related feedbacks in complex processes (Rosenfeld and Lensky, 1998; Kessler, 1969; Liu et al., 2006).

Active remote sensing systems, such as ground-based, satellite-borne, and airborne radars, provide precise vertical microphysical cloud data but are constrained by narrow observational swaths (Battaglia et al., 2020; Protat et al., 2009; Fox and Illingworth, 1997). In contrast, passive remote sensing enables large-scale cloud monitoring through multispectral measurements of reflected solar radiation and emitted thermal radiation, offering broad coverage with high spatiotemporal resolution (Nakajima et al., 2010a; Letu et al., 2020; Tana et al., 2023; Shi et al., 2025; Letu et al., 2023; Tang et al., 2025). However, conventional plane-parallel cloud assumptions in passive satellite cloud retrievals contradict natural three-dimensional cloud structures (Platnick, 2001; Horváth and Davies, 2004). Integrating passive and active satellite measurements can effectively combine their complementary advantages, thereby significantly enhancing the ability of passive sensors to retrieve cloud vertical microphysical properties.

Polarimetric multiangle imagers are widely regarded as pivotal instruments for acquiring multidimensional information in global and regional cloud property retrievals (Wang et al., 2022; Bréon and Goloub, 1998; Shang et al., 2019). Unlike conventional passive optical satellite payloads, these advanced sensors synergistically combine multiangle, multipolarization, and multispectral characterization capabilities, thereby maximizing observational information for individual pixels and specific targets (Dubovik et al., 2019). Moreover, the CER can be retrieved more reliably and robustly through polarized multiangle observation data, which also provides effective variance information, than through use of the MODIS imager (Breon and Doutriaux-Boucher, 2005; Bréon and Goloub, 1998; Shang et al., 2019). These factors significantly increase the potential for retrieving cloud vertical properties from passive satellite observations. The POLDER series represents the most mature polarimetric multiangle payloads internationally. Although POLDER-3, the final payload launched in 2004, was

decommissioned in 2013, the upcoming 3MI payload onboard the EUMETSAT Polar System Second Generation (EPS-SG) program will effectively inherit and improve POLDER-3's observational capabilities (Fougnie et al., 2018).

While significant research efforts have focused on cloud vertical structure retrieval (Rosenfeld and Lensky, 1998; Chen et al., 2020; Alexandrov et al., 2020; Barker et al., 2011; Leinonen et al., 2019; Shang et al., 2023), substantial challenges persist in large-scale vertical microphysical characterization, including insufficient prior knowledge from active sensors to guide passive retrieval algorithms and poorly understood correlations between key profile features and other cloud parameters. Additionally, comprehensive statistical analyses examining global-scale cloud profiles from structural and morphological perspectives are lacking. These gaps highlight the need for enhanced integration of active and passive systems with advanced and novel approaches in cloud profile retrieval. To address the above challenges, this paper focuses on quantifying and retrieving the vertical microphysical characteristics of single-layer stratiform liquid clouds. Using observation data from the CloudSat cloud profile radar (CPR), we extract global stratiform liquid cloud profile data over nearly three years. Through a novel perspective of analyzing profile shapes and structural characteristics, we aim to compensate for the lack of a priori knowledge of cloud profile retrieval by passive observation, to understand the significance of profile shape in the cloud life cycle, and to explore the correlation between structural features of the cloud profile and other cloud parameters. Moreover, this study reports the first retrieval of key cloud profile features from POLDER/Parasol satellite observations and the reconstruction of complete cloud profiles.

## 2 Datasets

## 2.1 CloudSat data

The main mission of the CloudSat satellite is to detect cloud vertical structures and improve the understanding of cloud abundance, distribution, structure and radiation characteristics. To date, the CloudSat official website has released observations from 2006 to 2020. The instrument carried by CloudSat is the Cloud Profile Radar (CPR), which is a 94-GHz millimeter-wave radar with a sensitivity 1,000 times that of a standard weather radar. The CPR transmits energy to the Earth and calculates the energy returned by the cloud as a function of distance. Global CER profiles of liquid stratiform clouds are derived from the latest version (R05) of CloudSat's 2B-CWC-RO product. This dataset provides vertical measurements of hydrometeor water content, number concentration, and effective radius, featuring 125 layers at 240-m vertical resolution across a 30-km detection range (Austin, 2007). Moreover, we employ the 2B\_CLDCLASS\_radar product (R05) of CloudSat for cloud layer information, including cloud layer type, cloud layer base height, cloud layer top height, land/sea flag, etc. The precipitation flags are obtained from the 2B-CLDCLASS product (R05) of CloudSat. Users can download all standard CloudSat products from the official CloudSat Data Processing Center at https://www.cloudsat.cira.colostate.edu.

As part of the A-train satellite constellation, CloudSat can also offer synergistic observation capabilities with other passive sensors, such as MODIS and POLDER. Although CloudSat/CPR is sensitive to hydrometers, uncertainties inevitably exist in these products: (1) Near the cloud base, the presence of drizzle or raindrops—due to predefined threshold settings—may hinder the ability to reliably distinguish them from cloud droplets, (2) contamination by drizzle and raindrops can cause deviations in the observed cloud droplet size distribution from the theoretical distribution, thereby introducing errors in the retrieval process (Austin, 2007), and (3) due to the limitations of 240-m resolution, it may not be possible to identify ultrathin layer structures below 240m.

## 2.2 Parasol data

100

115

The POLDER-3/Parasol payload is a multiangle, multipolarized, and multispectral instrument designed for atmospheric aerosol, cloud, water vapor, and radiation budget studies. Operating from September 2004 to October 2013, POLDER-3 features three polarized (490, 670, and 865 nm) and six nonpolarized (443, 565, 763, 765, 910, and 1020 nm) observation channels, providing atmospheric data from up to 16 angles with a nadir resolution of 5.3×6.2 km (Deschamps et al., 1994). Parasol joined the A-Train constellation in 2005, but regrettably, it drifted away from the formation in 2009. Nevertheless, the combined POLDER and CloudSat data still hold significant research value. This study employs their observations for the remote sensing of profiles, offering insights for future sensor combinations such as 3MI and EarthCARE.

Our study uses POLDER-3 Level-1 (L1\_B) products and cloud optical thickness parameters from Level-2 (RB2) products as input data for estimating the structural characteristics of cloud profiles. Moreover, we employ the CER retrieved by an improved primary cloudbow retrieval (PCR) algorithm (Shang et al., 2019) and the cloud-base height and cloud-top height retrieved based on POLDER data. The PCR algorithm permits an extended range of CER (3–25µm) and EV (0.01–0.29) estimates and a higher resolution (40–60 km) in the retrieval by using POLDER polarized measurements from both primary and supernumerary cloudbow regions. The retrieval algorithm for cloud base height will be thoroughly described in a forthcoming article and is therefore not discussed here

#### 120 3 Methodology

The study workflow (Fig. 1) is divided into three main parts: first, classifying the shapes of CER profiles from CloudSat data and examining their structural features; second, conducting correlation analyses between the structural features of these profiles and other relevant cloud parameters; and third, parameterizing cloud profiles, retrieving their key characteristics and reconstructing the complete profile. The methods employed in this study are introduced below.

Figure 1: Research Framework Flowchart (TP\_CER represents the cloud effective radius at the profile turning point; TP\_NCOT represents the normalized cloud optical thickness at the profile turning point; CT\_CER represents the cloud effective radius at the cloud top; CB\_CER represents the cloud effective radius at the cloud base; TP\_LWC represents the liquid water content at the profile turning point; TP\_NH represents the normalized height at the profile turning point; LWC represents the liquid water content; CTH represents the cloud-top height; CBH represents the cloud-base height; CGT represents the cloud geometric thickness.)

# 3.1 Cloud profile data preprocessing and shape simplification

To ensure the accuracy and consistency of the research data and to provide a reliable basis for subsequent analysis, we preprocessed the CloudSat profile data. Preprocessing involves mainly data screening and matching, as well as standardizing the data to better reveal the structural characteristics of the profiles. Due to the complexity of multilayered clouds and the dominance of single-layer cases, we limit our investigation to single-layer cloud profiles. Based on global CloudSat data product over nearly three years (2013, 2019, and the first eight months of 2020), we extracted approximately 12.47 million

CER profiles of single-layer stratocumulus and stratus clouds, with each profile linked to multiple auxiliary cloud parameters. The auxiliary information includes profile observation time, geographic coordinates, cloud-top height, cloud-base height, geometric thickness, liquid water path, corresponding liquid water content (LWC) profile for each CER profile, cloud type, land/sea flag, and precipitation flag. Moreover, the normalized optical thickness and normalized height at each layer of the profile are calculated for subsequent comparative analysis. Notably, the normalized optical thickness is 0 at the cloud top and 1 at the cloud base, whereas the normalized height is 1 at the cloud top and 0 at the cloud base. The selection of 2013, 2019, and 2020 is not arbitrary but strategically chosen to align with existing polarized multi-angle payload observations. Furthermore, since our study focuses on relatively homogeneous and stable single-layer stratiform liquid clouds, localized atmospheric anomalies do not impact the statistical results presented herein.

Here, we propose a shape simplification scheme primarily based on the Visvalingam-Whyatt line simplification (VM) algorithm (Visvalingam, 2016) to extract the essential shape characteristics of profiles. This approach effectively simplifies complex profile geometries while preserving their fundamental structure, thereby eliminating interference with the extraction of key profile features. The VM algorithm, which was originally designed for vector data processing, preserves geometric characteristics while reducing the number of data points. The simplification steps include (1) setting a distance threshold (decimation factor) as the simplification criterion; (2) calculating the area of triangles formed by consecutive points and identifying the smallest area (Amin); and (3) deleting the middle vertex if Amin 

Figure 2: Distribution and composition of effective data on the investigated liquid stratiform cloud profiles. (a) Geographical distribution of effective data; (b) statistical chart of land/sea and precipitation/nonprecipitation composition of effective data; (c) distribution and number of profile valid data bins.

Through an extensive literature review and visual analysis of CloudSat single-layer liquid cloud profiles, the vertical variation of cloud effective radius (CER) can be classified into four distinct shapes based on the monotonicity between adjacent layers: (1) triangle shaped (Inc\_Dec), increasing then decreasing; (2) monotonically decreasing (Mono\_Dec); (3) monotonically increasing (Mono\_Inc); and (4) decreasing then increasing (Dec\_Inc). These shapes depict the vertical variation in the CER from the cloud base to the top. The four shapes can be simply expressed by the following formulas:

$$Inc\_Dec: CER_1 < CER_2 < \dots < CER_k > CER_{k+1} > \dots > CER_N, 1 < k < N$$
(3)

Mono Dec: 
$$CER_1 > CER_2 > \dots > CER_N$$
 (4)

Mono\_Inc: 
$$CER_1 < CER_2 < \dots < CER_N$$
 (5)

$$Dec_{Inc:} CER_1 > CER_2 > \dots > CER_k 

Figure 3: Categorical statistics of stratiform cloud profile shape and the distribution of CER and normalized optical thickness at the TP of Inc\_Dec (Shape 1) profiles. (a) Shape of all profile data; (b) shape of land/sea profile data; (c) TP\_CER of stratiform cloud profiles; (d) TP\_NCOT of stratiform cloud profiles.

These shapes correspond to distinct development stages of the cloud life cycle (Fig. 4). Initial updrafts drive adiabatic growth, leading to a reduction in droplet size with increasing optical thickness. Mature clouds exhibit enhanced evaporation at the cloud top due to dry air entrainment, further reducing droplet size (Shape 1, Fig. 4(a)). In contrast, collision-coalescence near the cloud base promotes droplet growth and spectral broadening (Shape 2, Fig. 4(d)). The distinction between Fig. 4(c) and (d) lies in the presence of precipitation below the cloud base, which results in larger base particle sizes. If precipitation-induced droplet accumulation in the lower cloud layer reduces droplet size (based on Fig. 4(d)), the resulting pattern resembles that in Fig. 4(b).

Since the Inc\_Dec shape is more complex than the Mono\_Dec shape is and more structural characteristics require investigation (e.g., turning points of profiles), this study focuses specifically on analyzing the structural characteristics of Inc\_Dec profiles. The analysis establishes a foundation for subsequent feature parameterization and estimation. Unlike other shapes, the key parameters essential for describing Inc\_Dec profiles are the turning-point CER (TP\_CER) and the turning-point normalized optical thickness (TP\_NCOT). Fig. 3(c) shows the TP\_CER distribution for Inc\_Dec profiles.

Nonprecipitating stratiform clouds exhibit a single-peak TP\_CER distribution between 8 and 25μm, peaking near 12μm. Precipitating stratiform clouds exhibit a single-peak TP\_CER distribution between 10 and 25μm, peaking near 17μm. The analysis reveals several key results: (1) precipitating cloud profiles present larger (by 3–5μm) TP\_CERs than nonprecipitating clouds do, and (2) oceanic stratiform cloud profiles present a narrower TP\_CER range than their land counterparts do.

The distributions of TP\_NCOT for stratocumulus and stratus clouds are shown in Fig. 3(d). TP\_NCOT indicates the position of the TP within the cloud profile. The key observations include the following: (1) the TP\_NCOT of liquid stratiform clouds exhibits a multipeak distribution; (2) nonprecipitating cloud TPs are concentrated in the upper optical thickness region (NCOT: 0–0.5), while land-precipitating cloud TPs show a symmetric distribution around NCOT=0.5, and oceanic-precipitating cloud TPs are distributed predominantly between 0.5 and 1. This suggests that precipitating cloud TPs occur closer to the cloud base than nonprecipitating clouds do, which is consistent with the cloud life cycle illustrated in Fig. 3. (3) The peak of the oceanic precipitating cloud distribution clusters around NCOT=0.5.

Figure 4: Schematic diagram of the cloud life cycle stages corresponding to different profile shapes. (a) and (b) represent cloud droplet development corresponding to Inc\_Dec (triangle-shaped) profiles; (c) and (d) represent cloud droplet development corresponding to Mono Dec profiles.

## 4.2 Correlation analysis of profile structural features

During the profile retrieval process, to determine the main structure of the cloud profile, it is necessary to obtain the relevant information at the profile TP, as well as at the bottom of the cloud. Many previous studies have focused on the retrieval of cloud bottom parameters; however, few studies have explored cloud profile TP information, so we would like to establish a parameterized scheme and estimation method for cloud profile TP-related information. To further investigate the correlation between the structural features of the profile and other cloud parameters, we initially examined the relationships of both TP\_CER and TP\_NCOT with other cloud parameters for Inc\_Dec profiles. The study randomly selected 4800 data points for each of the eight stratiform clouds to be analyzed, and the correlations between the TP\_CERs of the eight stratiform clouds and the nine cloud parameters are presented in Fig. 5(a). For different types of clouds, TP\_CER has a strong correlation with cloud-base CER and the liquid water path (LWP), which generally range from 0.75 to 0.92, among which the cloud-base CER has the highest correlation with the TP\_CER of the stratiform cloud profiles. Moreover, TP\_CER also has a strong correlation with the liquid water content at the TP (TP LWC), and the correlation between stratiform cloud TP CER and

TP\_LWC fluctuates over a wider range, between 0.462 and 0.784. The cloud-base height, TP\_NH, and TP\_NCOT have weak correlations with TP\_CER, with correlations in the range of -0.26 to 0.13. The cloud-top CER and cloud-top height have different degrees of correlation for the TP\_CERs of different types of clouds: the correlation of the cloud-top CER is greater for the TP\_CER of nonprecipitating stratiform clouds (0.42–0.51) than that of precipitating stratiform clouds (0.23–0.27), and the correlation of cloud-top height with the TP\_CER of the sea stratocumulus and stratus profiles is greater (0.41–0.49) than that with the land stratiform clouds (0.18–0.29).

Figure 5: Correlation of TP parameters with other cloud parameters. (a) correlation of turning point CER with other cloud parameters; (b) correlation of the normalized optical thickness at turning point with other cloud parameters. The nine cloud parameters corresponding to the vertical axis are, from top to bottom: cloud top CER (CT\_CER), cloud bottom CER (CB\_CER), liquid water path (LWP), cloud top height (CTH), cloud bottom height (CBH), cloud geometric thickness (CGT), normalized height at the turning point (TP\_NH), normalized optical thickness at the turning point (TP\_NCOT), and liquid water content at the turning point (TP\_LWC). The nine types of clouds corresponding to the horizontal axis are, from left to right: land non-precipitation stratocumulus (Land\_Sc\_np), sea precipitation stratocumulus (Sea\_Sc\_np), land precipitation stratocumulus (Land\_St\_np), sea non-precipitation stratus (Land\_St\_np), sea non-precipitation stratus (Sea\_St\_np), land precipitation stratus (Land\_St\_np), sea non-precipitation stratus (Sea\_St\_np), land precipitation stratus (Land\_St\_np), sea precipitation stratus (Land\_St\_np), sea non-precipitation stratus (Sea\_St\_np), land precipitation stratus (Land\_St\_np), sea precipitation stratus (Sea\_St\_np).

Fig. 5(b) shows the correlations between the TP\_NCOT of stratiform cloud profiles and other cloud parameters. In marked contrast to TP\_CER, the correlations of almost all cloud parameters with TP\_NCOT, except for TP\_NH, are relatively weak, and almost all of them are in the range of  $\pm 0.3$ . Since both TP\_NH and TP\_NCOT indicate where the TP occurs in the cloud,

it is understandable why they are highly correlated. The weak correlation for TP\_NCOT stems from the fact that the TP position is largely independent of common cloud parameters such as droplet size, cloud water content, and cloud thickness. Instead, it is primarily influenced by microphysical processes like cloud-top entrainment and precipitation formation, leading to a relatively random distribution of TP\_NCOT within the cloud layer. This inherent randomness makes it inherently difficult to estimate TP\_NCOT using conventional correlation-based method.

To further investigate several parameters that are highly correlated with TP CER and their relative distributions with respect to TP CER, we randomly selected 4800 samples with four cloud characteristics (Land Nonprecip, Sea Nonprecip, Land Precip, and Sea Precip) to generate scatter density plots. The relative distributions of parameters highly correlated with the TP CER of stratiform cloud profiles are shown in Fig. 6(a)-6(p). As illustrated in Fig. 6(a)-(d), the scatter points are located mostly below and close to the 1:1 line, indicating a strong linear correlation between the CB CER and TP CER of the 355 profile. For sea cloud profiles, TP CER also shows a strong linear correlation with the LWP. In contrast, Fig. 6(e) and (g) show that, for land clouds, while TP CER still strongly correlates with the LWP, the scatter density plot exhibits a vertical distribution along TP CER, with points diverging from the density center toward both sides. Sea cloud scatter points, however, show a radial distribution pattern, dispersing symmetrically from the center. In Fig. 6(i)–(l), the scatter points form nearly horizontal stripes, with most points concentrated around several CGT values. This is due to CloudSat's measurement 360 method, which results in a discontinuous discrete distribution of CGT, thereby lowering the linear correlation between CGT and TP CER. TP LWC shows a high linear correlation with TP CER, with multiple density centers visible in the scatter density plots. As shown in Fig. 6(m), (n), and (p), nearly all plots exhibit two density centers. The two density centers observed in the relationship between the TP CER and TP LWC reflect two distinct cloud microphysical regimes. One is primarily driven by condensational growth, which tends to occur under low aerosol and stable conditions, resulting in higher 365 LWC for a given droplet size. The other is dominated by collision-coalescence, typical in relative high aerosol and dynamically active environments, leading to lower LWC for the same droplet size.

Figure 6: Relative distribution of cloud parameters (high correlation with TP\_CER) and profile TP\_CER. The green  $\times$  represents binned median of the parameter represented by the vertical axis; the black line segment represents binned median of the parameter represented by the vertical axis  $\pm \sigma$ .

# 4.3 Estimation of key structural features of CER profiles

Based on the previous analysis, CB\_CER and the LWP clearly have good linear correlations with TP\_CER. Therefore, we employ multiple linear regression to estimate TP\_CER for cloud profiles with four different characteristics using CB\_CER and the LWP as parameters. The goal is to derive empirical fitting formulas for TP\_CER based on these two parameters. Three combinations of dependent variables were selected: (1) CB\_CER and the LWP, (2) CB\_CER alone, and (3) the LWP alone. For all four cloud types, using the combination of CB\_CER and the LWP for multiple linear regression clearly yields the highest accuracy for estimating TP\_CER. The validation results of TP\_CER estimation, shown in Fig. 7, indicate that the RMSEs are 1.19 for Sea Nonprecip, 1.30 for Sea Precip, 1.75 for Land Nonprecip, and 1.96 for Land Precip. The estimation accuracy for sea cloud profiles is greater than that for land cloud profiles. Additionally, for sea cloud profiles, the use of both parameters for estimation significantly improves accuracy compared with the use of only one parameter. However, for land cloud profiles, the accuracy of TP\_CER estimation using only CB\_CER is comparable to the accuracy when both CB\_CER

and the LWP are used, suggesting that CB\_CER alone is sufficient for estimating TP\_CER for land cloud profiles. The final empirical fitting coefficient, predictive performance and more specific verification results are shown in the Appendix A.

Figure 7: Accuracy verification contour plots of the optimal TP\_CER estimation method for four different types of clouds (sea nonprecipitating, sea precipitating, land nonprecipitating, and land precipitating).

In addition to TP\_CER, we use random forest and multiple linear regression methods, considering various parameter combinations to estimate TP\_NCOT, according to the relatively weak correlation between other cloud parameters and TP\_NCOT. The results indicate that the random forest method performs well for estimating TP\_NCOT. The highest accuracy for estimating TP\_NCOT is achieved by combining four parameters—CB\_CER, CT\_CER, CGT, and the LWP—with RMSEs ranging from 0.1 to 0.12 and an R value of approximately 0.6, as shown in Fig. 8. Except for CTH, which can somewhat replace CGT for estimating TP\_NCOT, other parameter combinations significantly reduce estimation accuracy.

Figure 8: Accuracy verification contour plots of the optimal TP\_NCOT estimation method for four different types of clouds (sea nonprecipitating, sea precipitating, land nonprecipitating, and land precipitating).

This study utilizes POLDER-3 Levels 1 and 2 (RB2) data, with observations commencing on March 2, 2007, at 06:41:09, in conjunction with CloudSat products, to perform active-passive satellite data matching. The primary objective is to investigate the applicability of the proposed profile structure characterization method for retrieving cloud vertical structures from passive satellite data. This study focuses on a typical stratocumulus cloud region over the Indian Ocean, located west of Oceania, within the geographical range of 40°S-65°S and 100°E-125°E, as shown in Fig. 9(a). The cloud-base CER is difficult to obtain directly through satellite observations. Here, we first analyzed the statistical distribution of CloudSat-observed CB\_CER and found its probability density function to be highly regular. Based on this observation, we developed a multivariate regression model using known parameters (CTH, the LWP, CT\_CER, CBH) to estimate CB\_CER. The method achieved excellent results, with the highest retrieval accuracy (for sea nonprecipitating clouds) and an RMSE of 1.13 μm. Based on the aforementioned estimation method, we estimated the key structural features of the cloud field profile in Fig.

9(a)—TP\_CER and TP\_NCOT—with the results shown in Fig. 9(f). Here, we estimate only the TP information for profiles classified as Inc Dec.

Figure 9: Comparative results of estimating the TP using passive data (Parasol) and active data (CloudSat). (a) True-color image drawn by Parasol observation on March 2, 2007; (b)–(e) and (g)–(j) represent the 8 profile cases indicated in (a) and present a comparison between the TP estimated by Parasol data and those estimated by CloudSat data, as well as the profiles and TP observed by CloudSat; (f) shows the TP\_CER and TP\_NCOT results estimated by POLDER observation. The last row of panels in the figure represents the profiles reconstructed by CPRM; (k)–(m) and (n)–(p) correspond to cases 1 and 2, respectively.

The estimation accuracy is validated by comparing the estimated TP parameters derived from the POLDER-3/Parasol and CloudSat input parameters against the actual CloudSat measurements, as illustrated in Fig. 9(b)-(e) and (g)-(j). Eight representative profiles with TPs are selected based on CloudSat data. The comparative analysis reveals that the estimation accuracy when CloudSat parameters are used slightly surpasses that when POLDER-3 data are used. In particular, cases 3, 4,

5, 6, and 8 demonstrate satisfactory estimation performance, whereas cases 1, 2, and 7 exhibit relatively lower accuracy. This discrepancy can be attributed to three primary factors: (1) The coarse resolution of POLDER products cannot capture inherent subpixel heterogeneity. This study utilizes the POLDER Level 2 RB2 product (16 km resolution) along with cloud top height (CTH), cloud base height (CBH) (both at 6 km), and cloud top effective radius (CER) data (50 km resolution), all of which are derived from POLDER Level 1 products. Compared to CloudSat data, the coarser resolution of POLDER may cause biases resulting from subpixel heterogeneity. We conduct a further analysis of the eight cases in Fig. A2 by averaging the CloudSat CER profiles within each corresponding POLDER pixel along the altitude dimension. This process effectively aggregates the high-resolution CloudSat profiles to the spatial scale of a POLDER pixel, simulating what POLDER would likely "see". The resulting averaged profiles are then compared against our validation data—the CloudSat profile closest to the center of the POLDER pixel. Although this study specifically targets horizontally relatively homogeneous single-layer stratiform water clouds, subpixel heterogeneity—resulting from POLDER's coarse resolution—remains one of the main sources of error in estimating the structural parameters of cloud profiles. (2) Inherent physical limitation: Vertically integrated signal. The retrieval of CER from POLDER observations differs from traditional dual-channel methods, relying instead on the directional characteristics of polarized reflectance within the cloud bow scattering angle range. It should be emphasized that all such passive retrieval techniques essentially provide a vertically integrated measurement—a weighted average signal sensitive to microphysical properties from the cloud top downward through a depth determined by cloud optical thickness. This fundamental characteristic inherently increases the uncertainty in retrieving vertical structural features. (3) Propagation of input uncertainties. Errors in the upstream inputs (COT, CT CER from POLDER, and CTH/CBH from the combination of POLDER and ancillary data) inevitably propagate into the LWP and CB CER. This accumulated uncertainty then propagates into errors in the final estimated profile parameters, including the TP CER and its location. Cases with higher sub-pixel heterogeneity or where the cloud-top CER is less representative of the layer-average are particularly susceptible to this propagation effect.

# 445 5 Discussion and conclusion

The primary goal of this study was to analyze the structural features and shapes of single-layer stratiform liquid cloud CER profiles using global CloudSat data, with a focus on understanding how these profiles represent different stages of the cloud life cycle. We also aimed to retrieve key profile characteristics from multiangle passive imager observations and reconstruct complete cloud profiles using physical parameterization models.

Profile analysis of global single-layer stratiform liquid clouds reveals two dominant profile shapes: Inc\_Dec (triangular shape, 39.7%) and the Mono\_Dec (48.8%), which represent nearly 90% of cases. These shapes occur in both precipitating and nonprecipitating clouds, reflecting different lifecycle stages. For Inc\_Dec profiles, the turning-point (TP) CER (TP\_CER) and its position are structurally significant, showing strong correlations with cloud-base CER (CB\_CER), the liquid water

path (LWP), cloud geometric thickness (CGT), and liquid water content at the TP (TP\_LWC). In contrast, the normalized optical thickness at the TP (TP\_NCOT) depends primarily on its normalized height (TP\_NH) and weakly on other parameters.

Multilinear regression is applied using POLDER-3 data to estimate TP\_CER. For maritime clouds, combining CB\_CER and the LWP achieves high accuracy (RMSE: 1.19–1.30μm), whereas continental clouds require only CB\_CER (RMSE: 1.75–1.96μm). For TP\_NCOT, random forest outperforms linear regression, with optimal results (RMSE≈0.1) using CB\_CER, cloud-top CER (CT\_CER), CGT, and the LWP. Cloud-top height (CTH) could partially substitute for CGT.

The primary challenges in retrieving profile structural features originate from the following aspects: (1) The coarse resolution of POLDER products restricts the ability to capture sub-pixel cloud heterogeneity; however, by concentrating on relatively uniform single-layer stratiform liquid clouds, this study partially mitigates the resulting retrieval uncertainties. It should be noted that sub-pixel heterogeneity can inevitably introduce certain errors, particularly at cloud boundaries. Nevertheless, Shang et al. (2015) pointed out that the error caused by sub-pixel heterogeneity in cloud effective radius (CER) retrieval does not exceed 10%, which remains within an acceptable range. (2) The estimation of CB\_CER remains subject to certain uncertainties due to the inherent challenges in retrieving cloud base microphysical properties from passive observations; (3) The 240-m vertical resolution of CloudSat is insufficient to resolve ultra-thin cloud layers or capture fine-scale in-cloud structures, such as sharp inversion layers or thin drizzling layers near cloud base.

To address the issues mentioned above, the improvement strategies below can be implemented: (1) Internationally, there are currently polarimetric multi-angle payloads with higher spatial resolution and greater observation angles that have been launched or are planned for launch. For instance, China's DPC/GF-5 achieves a spatial resolution of nadir 3.3 km; the 3MI/Metop-SG developed by the European Space Agency offers a spatial resolution of nadir 4 km, supports up to 21 observation angles, and incorporates near-infrared bands. These capabilities collectively enable higher-resolution CER retrieval. (2) In terms of estimating the CB\_CER: a) Introduce meteorological factors—such as ERA5 reanalysis data (e.g., wind speed, temperature, humidity, pressure, vertical velocity)—to assist in estimating the CB\_CER, thereby enhancing the physical characterization of the cloud-base environment; b) Optimize existing methods for directly retrieving cloud-bottom particle size using passive observations (Level 1 or Level 2 products), improve the robustness of the retrieval model, and clarify the applicable boundaries of the method—specifically, determining the optical thickness threshold beyond which passive observations can no longer capture cloud-bottom information; c) Incorporate an uncertainty weighting framework to dynamically adjust the contribution weights of different input parameters based on their reliability, thereby refining the retrieval accuracy of CB\_CER and reducing dependence on CloudSat-derived empirical relationships. (3) For the issue that CloudSat is insufficient to distinguish thin clouds or fine cloud structures smaller than 240m, in the future, the EarthCARE/CPR observation data with higher vertical resolution can be adopted to alleviate this problem. EarthCARE/CPR

demonstrates notable advancements over CloudSat, most significantly through its finer vertical resolution of 100 m compared to CloudSat's 240 m. Additional enhancements include higher detection sensitivity, Doppler-based vertical wind measurements, and synchronized multi-sensor observational capabilities. These improvements are expected to deliver enhanced observational capabilities for characterizing finer-scale cloud microphysical processes and their interactions with atmospheric dynamics.

515

Meanwhile, the validation results indicate that the RMSE of stratiform cloud profile structural characteristics over land is significantly higher than that over sea. This discrepancy is considered to be mainly attributable to the following factors: a) Sub-pixel Surface Heterogeneity: Variations in surface reflectance among different land cover types (e.g., vegetation, bare soil, urban areas) lead to mixed-pixel effects, complicating the decoupling of cloud optical properties. b) Aerosol Interference: Higher and spatiotemporally variable aerosol loadings over land can perturb cloud signals either indirectly by 500 altering cloud microphysics (e.g., through cloud condensation nuclei effects) or directly via scattering, c) Surface Heating Effects: The lower thermal inertia of land surfaces results in more complex boundary-layer dynamics, increasing spatiotemporal variability in cloud base height and cloud layer thickness, which in turn elevates retrieval uncertainty. d) Interference from complex terrain and high-albedo surfaces: Complex terrain (e.g., mountains) and high-albedo surfaces (e.g., snow cover) are prone to causing false positives in cloud detection or overestimation of optical thickness. It is suggested that 505 the following strategies could be adopted in the future to improve the estimation accuracy of stratiform cloud profile structural characteristics over land: a) Integration of land cover classification data (e.g., MODIS Land Cover product); b) Integration of aerosol ancillary data: Multi-source aerosol observations (e.g., MERRA-2 reanalysis data, AERONET groundbased measurements) could be incorporated to better constrain retrieval parameters in regions affected by aerosol-cloud interactions; c) Development of advanced retrieval algorithms: More sophisticated methods, such as machine learning or 510 deep learning approaches, could be employed to better represent the complex relationships between land surface, atmosphere, and clouds.

Future work should focus on higher-resolution observations and improved retrieval methods to refine cloud structural analysis. In summary, this study advances methods for estimating TP characteristics in liquid clouds but underscores the need for enhanced observational capabilities and hybrid active-passive approaches to fully resolve profile uncertainties. Additionally, our work on the parameterization and retrieval of liquid cloud profiles through multiangle passive imagers provides valuable insights that can further improve the understanding and modeling of cloud processes in weather and climate systems.

# 520 Appendix A

Table A1. Table of acronyms.

|              | •                                                                  |
|--------------|--------------------------------------------------------------------|
| Abbreviation | Full term                                                          |
| CALIPSO      | Cloud-Aerosol Lidar and Infrared Pathfinder Satellite Observations |
| CB_CER       | Effective Radius of Cloud Base                                     |
| CBH          | Cloud Base Height                                                  |
| CER          | Cloud Effective Radius                                             |
| CGT          | Cloud Geometric Thickness                                          |
| CNES         | Centre National d'Études Spatiales                                 |
| CPM          | Cloud Profile Model                                                |
| CPR          | Cloud Profile Radar                                                |
| CPRM         | Cloud Profile Reconstruction Model                                 |
| CSU          | The Colorado State University                                      |
| CT_CER       | Effective Radius of Cloud Top                                      |
| CTH          | Cloud Top Height                                                   |
| Dec_Inc      | Decreasing then increasing                                         |
| ECMWF        | European Centre for Medium-Range Weather Forecasts                 |
| GCM          | General Circulation Model                                          |
| Inc_Dec      | Increasing then Decreasing                                         |
| LWP          | Liquid Water Path                                                  |
| MLR          | Multiple Linear Regression                                         |
| Mono_Dec     | Monotonically Decreasing                                           |
| Mono_Inc     | Monotonically Increasing                                           |
| POLDER       | Polarization and Directionality of Earth's Reflectance             |
| RAMS         | Regional Atmospheric Modeling System                               |
| RB2          | Level 2 products of POLDER-3                                       |
| RF           | Random Forest                                                      |
| RMSE         | Root Mean Square Error                                             |
| TP           | Turning Point                                                      |
| TP_CER       | Cloud Effective Radius at the Turning Point                        |
| TP_LWC       | Liquid Water Content at the Turning Point                          |
| TP_NCOT      | Normalized Cloud Optical Thickness at the Turning Point            |
| TP_NH        | Normalized Height at the Turning Point                             |

| 525 method | l |
|------------|---|
|            |   |

| Cloud profile type | E       | Empirical coefficient |                |       | Predictive performance |      |  |
|--------------------|---------|-----------------------|----------------|-------|------------------------|------|--|
| Cloud proffic type | $eta_0$ | $\beta_1(CB\_CER)$    | $\beta_2(LWP)$ | $R^2$ | R                      | RMSE |  |
| Sea Nonprecip      | 2.2656  | 0.8342                | 0.0052         | 0.77  | 0.90                   | 1.19 |  |
| Sea Precip         | 3.6904  | 0.7920                | 0.0022         | 0.87  | 0.94                   | 1.30 |  |
| Land Nonprecip     | 0.5844  | 1.1234                |                | 0.54  | 0.83                   | 1.76 |  |
| Land Precip        | 3.7843  | 0.8985                |                | 0.63  | 0.86                   | 1.96 |  |

**Table A3.** Statistics on the number of stratiform clouds categorized by land and sea, precipitating and nonprecipitating.

|                | Sc       | St     | Sc+ St   |
|----------------|----------|--------|----------|
| Land Nonprecip | 1034462  | 15823  | 1050285  |
| Sea Nonprecip  | 4016879  | 135926 | 4152805  |
| Land Precip    | 853197   | 28468  | 881665   |
| Sea Precip     | 6188612  | 199916 | 6388528  |
| Sum            | 12093150 | 380133 | 12473283 |

**Table A4.** Further statistics on CER profiles of complex shapes ("Other": category V in profile shape).

|                                        | Sc+            | -St           | Sc+St       |            |  |
|----------------------------------------|----------------|---------------|-------------|------------|--|
|                                        | Land Nonprecip | Sea Nonprecip | Land Precip | Sea Precip |  |
| Sum total                              | 137699         | 308344        | 84026       | 608258     |  |
| Percentage of situation 1 <sup>a</sup> | 33.57%         | 33.57%        | 47.88%      | 49.68%     |  |
| Percentage of situation 2 <sup>b</sup> | 31.41%         | 36.72%        | 19.07%      | 18.93%     |  |
| Intersection of 1 and 2°               | 8.40%          | 11.72%        | 6.31%       | 6.21%      |  |
| Percentage of situation 1+2            | 56.58%         | 58.57%        | 60.63%      | 62.40%     |  |

<sup>&</sup>lt;sup>a</sup> situation1 refers to a situation where only one segment of the profile does not correspond to the increasing and then decreasing shape profile of shape1; <sup>b</sup> situation2 refers to a situation where only one segment of the profile does not correspond to the monotonically decreasing shape profile of shape2. <sup>c</sup> There is an intersection of situation1 and situation2, i.e., a profile that matches both situation1 and situation2 (Intersection of 1+2), which needs to be subtracted out when calculating the sum of the two in order to avoid double counting.

**Table A5.** Validation accuracy of TP\_CER linear regression when different parameters are used.

| Independent variables  | Dependent variables | $\mathbb{R}^2$ | R    | RMSE | Regression<br>data<br>volume | Validation<br>data<br>Volume |
|------------------------|---------------------|----------------|------|------|------------------------------|------------------------------|
| TP CER                 | CB_CER;<br>LWP      | 0.77           | 0.90 | 1.19 | 1.400000                     | 1200520                      |
| (Sea Nonprecip)        | CB_CER              | 0.52           | 0.82 | 1.59 | 1400000                      | 1389529                      |
|                        | LWP                 | 0.61           | 0.85 | 1.47 |                              |                              |
| TP_CER<br>(Sea Precip) | CB_CER;<br>LWP      | 0.87           | 0.94 | 1.30 | 700000                       | 710750                       |
|                        | CB_CER              | 0.82           | 0.92 | 1.46 | 700000                       |                              |
|                        | LWP                 | 0.35           | 0.78 | 2.37 |                              |                              |
| TP_CER                 | CB_CER;<br>LWP      | 0.54           | 0.83 | 1.75 | 250000                       | 260000                       |
| (Land                  | CB_CER              | 0.54           | 0.83 | 1.76 | 250000                       | 260999                       |
| Nonprecip)             | LWP                 | 0.28           | 0.76 | 2.43 |                              |                              |
| TP_CER (Land Precip)   | CB_CER;<br>LWP      | 0.63           | 0.86 | 1.96 | 120000                       | 118706                       |

**Table A6.** Validation accuracy of TP\_NCOT predictions when different parameters and methods are used.

| Independent variables                                                                                      | Dependent variables      | R    | RMSE | rRMSE | Method |
|------------------------------------------------------------------------------------------------------------|--------------------------|------|------|-------|--------|
| TP_NormCOT (Sea Nonprecip)  TP_NormCOT (Sea Precip)  TP_NormCOT (Land Nonprecip)  TP_NormCOT (Land Precip) | CB_CER, CT_CER, CGT, LWP | 0.59 | 0.10 | 0.23  |        |
|                                                                                                            | CB_CER, CT_CER, CTH, LWP | 0.56 | 0.10 | 0.23  | RF     |
|                                                                                                            | CT_CER, CGT, LWP         | 0.39 | 0.12 | 0.27  | Kľ     |
|                                                                                                            | CB_CER, CT_CER, LWP      | 0.49 | 0.11 | 0.25  |        |
|                                                                                                            | CB_CER, CT_CER, CGT, LWP | 0.51 | 0.11 | 0.24  | MLR    |
|                                                                                                            | CB_CER, CT_CER, CGT, LWP | 0.63 | 0.11 | 0.22  |        |
|                                                                                                            | CB_CER, CT_CER, CTH, LWP | 0.64 | 0.11 | 0.21  | RF     |
|                                                                                                            | CT_CER, CTH, LWP         | 0.40 | 0.14 | 0.26  |        |
|                                                                                                            | CB_CER, CT_CER, CGT, LWP | 0.52 | 0.12 | 0.24  | MLR    |
|                                                                                                            | CB_CER, CT_CER, CGT, LWP | 0.58 | 0.12 | 0.29  |        |
|                                                                                                            | CB_CER, CT_CER, CTH, LWP | 0.53 | 0.13 | 0.30  | RF     |
|                                                                                                            | CT_CER, CTH, LWP         | 0.37 | 0.14 | 0.33  |        |
|                                                                                                            | CB_CER, CT_CER, CGT, LWP | 0.54 | 0.12 | 0.29  | MLR    |
|                                                                                                            | CB_CER, CT_CER, CGT, LWP | 0.66 | 0.12 | 0.27  |        |
|                                                                                                            | CB_CER, CT_CER, CTH, LWP | 0.65 | 0.12 | 0.28  | RF     |
|                                                                                                            | CT_CER, CGT, LWP         | 0.33 | 0.16 | 0.35  |        |
|                                                                                                            | CB_CER, CT_CER, CGT, LWP | 0.60 | 0.13 | 0.29  | MLR    |

Figure A1. Complex CER profile shapes and their possible corresponding simplified shapes (examples).

**Figure A2.** Comparison of the average CER profile (averaged by height) within the same POLDER pixel versus the CER profile closest to the center of the POLDER pixel.

Data Availability. All datasets used in this work are open-source. The CloudSat datasets are available from the CloudSat Data Processing Center of the Cooperative Institute for Research in the Atmosphere (http://www.cloudsat.cira.colostate.edu/). The Parasol products are available from ICARE Data and Services Center (https://www.icare.univ-lille.fr/).

**Author contribution.** HS and HL outlined the project, HS and YW conceived the methodology and performed the experiments, YW composed the manuscript, and all authors revised the manuscript. HS and HL funded, supervised, and encouraged the research.

Competing interests. The contact author has declared that none of the authors has any competing interests.

Acknowledgements. This work was supported by the National Key Research and Development Program of China (Grant No. 2023YFB3905900) and National Natural Science Foundation of China (B) (Grant No. 42522507). We are grateful to the Data Processing Center (DPC) managed by the Cooperative Institute for Research in the Atmosphere (CIRA) for providing CloudSat data, which has greatly facilitated our research. The authors would also like to thank ICARE Data and Services Center, Villeneuve d'Ascq Cedex, France, for providing the POLDER-3/Parasol product (https://www.icare.univ-lille.fr/).

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
