# Peer review of "Characterization of liquid cloud profiles using global collocated active radar and passive polarimetric cloud measurements"

_EGUsphere, 2025_

## Referee Comment (RC1)

**Review of "Characterization of liquid cloud profiles using global collocated active radar and passive polarimetric cloud measurements" by Wang et al.**

**General Summary**

This paper presents an innovative methodology for characterizing vertical profiles of stratiform liquid clouds. The authors identify dominant morphological patterns of cloud effective radius profiles using CloudSat radar data, and then develop a way to retrieve profile information from passive polarimetric (POLDER) satellite observations. The paper is technically strong and well-written. However, after reading the paper I was left with a few key questions that should be addressed before publication.

**Major Comments**

1) The authors state that cloud-base height is retrieved "based on POLDER data" (line 116). How, exactly, is this retrieval performed? I was not aware that cloud base height could be retrieved from POLDER. Does the multivariate regression model mentioned in line 367 also use cloud base height from POLDER?

2) The spatial resolution of CWC-RO (less than 2km) in vastly different from the spatial resolution of POLDER (~50km). I'd like to see more details about how the observations were matched up in creating Figure 9, and more discussion about whether relationships between variables derived at CloudSat resolution should be expected to hold at POLDER resolution, when there will be a lot of sub-pixel heterogeneity.

**Minor Comments**

Line 136: Why 2013, 2019, and the first eight months of 2020? This seems like a very arbitrary group of years to use.

Line 204: As far as I am aware, there is no "Colorado State University regional climate model." Do you mean the CSU Regional Atmospheric Modeling System (RAMS)?

Line 223: It should be noted that the CloudSat CWC-RO product misses many (perhaps the majority of) single-layer liquid clouds, either because the clouds are masked by surface clutter or because they are below the radar's noise threshold (e.g., Lamar et al., 2020; Schulte et al., 2023). So the true nonprecipitating-to-precipitating ratio is likely much higher.

> Lamer, K., Kollias, P., Battaglia, A., and Preval, S.: Mind the gap – Part 1: Accurately locating warm marine boundary layer clouds and precipitation using spaceborne radars, Atmos. Meas. Tech., 13, 2363–2379, https://doi.org/10.5194/amt-13-2363-2020, 2020.

> Schulte, R. M., Lebsock, M. D., and Haynes, J. M.: What CloudSat cannot see: liquid water content profiles inferred from MODIS and CALIOP observations, Atmos. Meas. Tech., 16, 3531–3546, https://doi.org/10.5194/amt-16-3531-2023, 2023.

Line 240: I believe you mean Table A4 here, but even so, I do not understand what the table is intended to show.

Line 327: Any idea whether these two density centers have physical meaning?

---

## Referee Comment (RC2)

**Review of "Characterization of liquid cloud profiles using global collocated active radar and passive polarimetric cloud measurements" by Wang et al.**

**Overall Comment:**

This study presents a significant methodological advancement in remote sensing of liquid cloud vertical profiles by integrating active (CloudSat radar) and passive (POLDER polarimetric imager) measurements . The work is rigorous and innovative, leveraging global-scale data (12.47M profiles) to classify cloud profile shapes, establish statistical correlations, and develop retrieval models. The paper does several things exceptionally well: 1. Global-scale profile classification is statistically robust, providing a comprehensive understanding of dominant cloud structures (triangle-shaped and monotonically decreasing profiles). 2. Shape simplification using the VM algorithm is clever, effectively reducing noise while preserving key structural features. 3. Correlation analyses (e.g., between TP_CER and CB_CER, LWP) provide valuable physical insights into cloud microphysics. 4. The hybrid ML/physical modeling approach (CPRM reconstruction) is methodologically sound, combining the strengths of data-driven and physics-based methods.

However, passive retrieval limitations and dependence on CloudSat for prior knowledge (especially CB_CER estimation) remain fundamental constraints. Current methods cannot fully resolve vertical uncertainties due to: 1. CloudSat's 240-m vertical resolution , which restricts layer precision (>2.64 km); 2. POLDER's coarse spatial resolution (50 km for CER) , which may misrepresent fine-scale cloud variations.

Despite these limitations, this work represents a significant step toward

parameterizing cloud processes in climate models , particularly for stratiform/stratocumulus clouds.

Major Comments :

1. The study's heavy reliance on CloudSat-derived empirical relationships for estimating CB_CER represents a critical constraint that significantly impacts the broader applicability of the methodology. While the multivariate regression approach using CT_CER, LWP, and CTH demonstrates reasonable correlation (0.75-0.92), this dependence on active sensor data fundamentally undermines the potential for truly independent passive retrievals. The propagation of errors through this empirical relationship is particularly concerning, with RMSE values reaching up to 1.96 μm for land-based clouds. This limitation is especially problematic because CB_CER serves as a foundational parameter for estimating TP_CER, meaning any errors in the initial cloud-base estimation will cascade through the entire retrieval process. The authors should more thoroughly discuss potential mitigation strategies, such as incorporating ancillary data sources or developing physics-based approaches to reduce this critical dependency on CloudSat for prior knowledge.

2. The observed inconsistencies in POLDER-based TP_CER and TP_NCOT retrievals, particularly evident in Cases 1, 2, and 7 of Figure 9, reveal important limitations in current passive sensing capabilities. These discrepancies likely stem from multiple compounding factors that warrant deeper examination. First, POLDER's relatively coarse spatial resolution (approximately 50 km for CER retrievals) means it cannot resolve fine-scale heterogeneities in cloud microphysical properties that CloudSat can detect along its narrow swath. Second, the visible-band measurements used by POLDER are primarily sensitive to cloud-top properties, making it inherently challenging to accurately characterize vertical microphysical gradients. The study would benefit from a more detailed error

analysis quantifying how these sensor limitations translate to uncertainties in profile reconstruction, perhaps through sensitivity studies or comparison with higher-resolution datasets where available.

3. CloudSat's 240-m vertical resolution, while impressive for spaceborne radar, imposes significant limitations on the study's ability to characterize thin or finely structured cloud layers. This resolution threshold means the CPRM reconstruction cannot resolve features smaller than about 2.64 km in vertical extent, potentially missing important microphysical transitions in shallow cloud systems. The impact is particularly relevant for stratocumulus clouds, which often exhibit thin but meteorologically important structures like sharp inversion layers or thin drizzling layers near cloud base. The authors should expand their discussion of how this resolution limitation affects the physical interpretation of their results, and perhaps suggest how future sensors with finer vertical resolution (like EarthCARE's radar) might overcome this constraint.

4. The coarse resolution of POLDER observations presents multiple challenges that extend beyond the immediate retrieval accuracy issues. At 50 km resolution, individual POLDER pixels often or possiblely integrate across multiple cloud regimes, potentially blending fundamentally different cloud types and obscuring important spatial gradients. This becomes particularly problematic when trying to apply the methodology to broken cloud fields or cloud edges, where sub-pixel variability is high. While the current focus on stratiform clouds is understandable given their relative homogeneity, the paper would benefit from a more thorough discussion of how partial cloudiness and three-dimensional radiative effects might bias the retrievals. The authors might consider adding a sensitivity analysis or at least a more detailed qualitative discussion of these effects in the limitations section.

5. The exclusion of 9.1% of profiles classified as complex-shaped introduces a

subtle but potentially important selection bias in the results. While this filtering improves the clarity of the statistical relationships, it risks creating an overly idealized representation of real-world cloud profiles. Many meteorologically significant situations - such as clouds undergoing strong entrainment, multilayered structures, or precipitating systems - may fall into this excluded category. The authors should more thoroughly justify their exclusion criteria and discuss how this might affect the generalizability of their findings. A sensitivity analysis showing how including some portion of these complex profiles affects the retrieval statistics would significantly strengthen the paper's conclusions.

6. The minimal three-year overlap between CloudSat (2006-2020) and POLDER (2004-2013) operations raises important questions about the dataset's representativeness for global climatological studies. Cloud properties exhibit significant interannual variability influenced by large-scale modes like ENSO, and a three-year period may not adequately capture this natural variability. The authors should discuss whether their training dataset (2013, 2019, and part of 2020) is truly representative of global cloud conditions, and whether the limited temporal sampling might introduce biases in the derived statistical relationships. This is particularly relevant given that some of the training years (like 2019) were characterized by unusual atmospheric conditions in certain regions.

7. The notably weak correlations for TP_NCOT (Figure 5b) reveal fundamental challenges in passively retrieving information about vertical structure inflection points. This poor correlation performance suggests that current passive observables may lack the necessary information content to reliably determine the normalized optical thickness at turning points. The authors should expand their discussion of potential physical reasons for this limitation, such as: The insensitivity of passive measurements to vertical redistribution of cloud water; The degeneracy between different vertical configurations that produce similar top-of-atmosphere signals;

The potential for cloud inhomogeneity effects to obscure the true profile characteristicsA more thorough exploration of these physical limitations would help readers better understand the boundaries of what can realistically be achieved with passive profile retrievals.

8. The substantially higher errors over land (RMSE 1.96 μm vs. ~1.3 μm over ocean) point to important unresolved challenges in land cloud retrievals that deserve more detailed discussion. Several factors likely contribute to this performance gap:

- Greater sub-pixel heterogeneity over land due to surface variability
- Higher and more variable aerosol loading affecting the cloud microphysics
- Stronger surface heating effects on cloud boundary layer dynamics
- Potential artifacts from the underlying terrain elevation and albedo

The paper would benefit from a dedicated discussion of these land-specific challenges and potential strategies to mitigate them, such as incorporating land surface type classifications or aerosol information into the retrieval framework.

Minor Comments:

1. Vertical resolution impact on thin layers:

While CloudSat's 240-m resolution is mentioned (Sec. 2.1), its inability to resolve sub-240 m layers (e.g., thin stratus) should be explicitly discussed.

Terminology consistency:

Line 218: "Stratiform water cloud profiles" → "liquid cloud profiles" for consistency with the rest of the paper.

---

## Author Comment (AC2)

Dear referees,

Thank you for your comments concerning our manuscript entitled *"Characterization of liquid cloud profiles using global collocated active radar and passive polarimetric cloud measurements"* (ID: *egusphere-2025-2471)*. Those comments are all valuable and very helpful for revising and improving our paper, as well as the important guiding significance to our researches. We have studied comments carefully and have made correction which we hope meet with approval. Revised portions are marked in blue (referee #1) /orange (referee #2) in our response document. The relevant references are at the end of our reply letter. The main corrections in the paper and the responses to the referees' comments are as following:

**Response to Referee #1's Comments**

**General Summary**

This paper presents an innovative methodology for characterizing vertical profiles of stratiform liquid clouds. The authors identify dominant morphological patterns of cloud effective radius profiles using CloudSat radar data, and then develop a way to retrieve profile information from passive polarimetric (POLDER) satellite observations. The paper is technically strong and well written. However, after reading the paper I was left with a few key questions that should be addressed before publication

**Major Comment 1:** The authors state that cloud-base height is retrieved "based on POLDER data" (line 116). How, exactly, is this retrieval performed? I was not aware that cloud base height could be retrieved from POLDER. Does the multivariate regression model mentioned in line 367 also use cloud base height from POLDER?

> **Response:** Thank you for your question. Obtaining cloud bottom heights based on POLDER data is another work in progress by the authors associated with this paper, which has been completed but not formally published, and is currently being submitted to relevant academic journals for review, so we do not describe this work in detail in this manuscript, and the multivariate linear regression model mentioned in line 367 in this study also uses the cloud bottom heights inverted by this method as input. We describe here the implementation of the method to obtain cloud bottom heights based on POLDER data to answer the questions raised by the referee:
>
> We developed a machine learning-based approach to estimate cloud base height (CBH) from POLDER/Parasol observations by leveraging collocated CloudSat radar measurements. The dataset was constructed by matching Parasol Level 1 (L1) data—including oxygen absorption (OA) channels (763 nm and 765 nm), OA ratios across 14 viewing angles, longitude, latitude, and elevation—with CloudSat Level 2 (L2) CBH products for March, June, September, and December 2007. Spatial collocation accuracy was constrained to within 0.01°, while temporal discrepancies were negligible due to the near-simultaneous observations from A-Train satellites. To ensure high-quality training data, only cases where Parasol confidently detected cloudy scenes and CloudSat identified single-layer clouds were retained. The dataset was split into training and validation subsets, with 7 days per month reserved for independent evaluation. The machine learning model used geographic coordinates (longitude, latitude, elevation) and

Parasol's multi-angle OA information as inputs to predict CBH, with CloudSat-derived heights serving as ground truth. After optimization and validation, the finalized model enabled global CBH retrieval using Parasol L1 data alone, providing a novel solution for passive sensor-based cloud vertical structure characterization. This method addresses the inherent limitations of passive remote sensing in directly probing cloud boundaries while capitalizing on POLDER's unique multi-angle OA capabilities. After several machine learning algorithms are compared, the deep neural network (DNN) model with the best accuracy is selected as the retrieval model. The method of CBH reversal based on multiangle OA remote sensing and the DNN has a mean absolute error (MAE) of 0.78 km, a bias of 0.22 km, and a correlation coefficient (R) of 0.82. By integrating machine learning with the multiangle OA, this method offers a novel approach for CBH retrieval. Fig. R1 shows the specific process of our CBH retrieval algorithm.

[Figure]

**Figure R1.** Flowchart of the machine learning-based algorithm for retrieving CBHs using the OA (Ji et al. 2025. Manuscript submitted for publication)

**Detailed modifications are as follows:** We have added the explanation to Line 118-119: "The retrieval algorithm for cloud base height will be thoroughly described in a forthcoming article and is therefore not discussed here."

**Major Comment 2:** The spatial resolution of CWC-RO (less than 2km) in vastly different from the spatial resolution of POLDER (~50km). I'd like to see more details about how the observations were matched up in creating Figure 9, and more discussion about whether relationships between variables derived at CloudSat resolution should be expected to hold at POLDER resolution, when there will be a lot of sub-pixel heterogeneity.

**Response:** Thank you very much for your suggestions. The issues you raised are crucial and will be essential for us to enhance the completeness of the paper and demonstrate the robustness of the estimated method for cloud-top profile structural characteristics.

(1) First, addressing your initial question: The purpose of the experiment in Fig. 9 is to estimate the profile turning point CER (TP_CER) and normalized optical thickness (TP_NCOT) using passive data, and to compare these results with active data. This aims to explore the feasibility of the aforementioned method for estimating profile characteristics in passive data. The matching process is as follows: We began by identifying matching pairs of POLDER3 and CloudSat data from March 2007, focusing on orbits that contained both datasets and included a high number of stratiform cloud profiles with a triangular shape. From these, we selected the POLDER3 data recorded between 06:41:09 and 07:24:06 on March 2, 2007, along with the corresponding CloudSat data, to estimate and validate the profile TP parameters.

We primarily used POLDER Level 2 data (RB2) for matching with CloudSat. The spatial resolution of POLDER RB2 product is approximately 16 km, which differs from CloudSat's spatial resolution (less than 2 km). Therefore, during the matching process, we calculated the Euclidean distance between each POLDER_RB2 pixel and the corresponding CloudSat data point. Due to the coarser resolution of POLDER_RB2 data compared to CloudSat, multiple CloudSat data points may correspond to the same POLDER_RB2 pixel. We retained only the CloudSat data point closest to the center of the POLDER_RB2 pixel. Ultimately, eight cases were randomly selected (Fig. 9) for validating the estimated characteristics of the profile structures.

Through matching, we extracted COT, latitude, longitude and other related data from the POLDER_RB2 product. Using these coordinates, we further extracted CBH, CTH, and CT_CER data retrieved by our algorithm. CBH and CTH were retrieved from POLDER3 Level 1 products with a resolution of 6 km, matching the L1 product resolution. CT_CER was also retrieved from POLDER L1 product with a 50 km resolution. POLDER lacks near-infrared bands, so it can only retrieve cloud-top CER using multi-angle polarization signals. This method has a drawback: it must compensate for insufficient angular sampling by including more pixels—resulting in lower resolution for the retrieved CER. Our current algorithm (Shang et al., 2019) can achieve CER retrieval at a range of 40－60 km. This paper utilizes the retrieved 50 km resolution CER product. However, we do not consider this an insurmountable permanent flaw. Internationally, there are currently polarimetric multi-angle payloads with higher spatial resolution and greater observation angles that have been launched or are planned for launch. For instance, China's DPC/GF-5 achieves a spatial resolution of nadir 3.3 km; the 3MI/Metop-SG developed by the European Space Agency offers a spatial resolution of nadir 4 km, supports up to 21 observation angles, and incorporates near-infrared bands. These capabilities collectively enable higher-resolution CER retrieval.

(2) We understand your concerns regarding sub-pixel heterogeneity due to the coarse resolution of POLDER data, as well as the challenges in applying relationships derived from CloudSat to POLDER data because of differing spatial resolutions. Our primary response is as follows: (a)

Our primary research subject is single-layer stratiform liquid clouds (stratocumulus and stratus). Relevant literature indicates (Jr., 2014) that within stratiform cloud regions, both updrafts and downdrafts are relatively weak. They are relatively uniform horizontally compared to other cloud types, and their cloud microphysical properties exhibit slow horizontal variations—that is, they are less spatially heterogeneous. This is why we selected single-layer stratiform liquid clouds—a structurally simpler cloud type—as our primary research subject. (b) Shang et al. (2015) specifically investigated the impact of liquid cloud spatial heterogeneity on CER retrieved from POLDER. The Table 2(Fig. R2) presented in their paper shows that under sub-grid scale heterogeneity, the relative deviation between the retrieved CER and the sub-grid scale CER mean ranges from 0.86% to 8.33% (Table R1), with none exceeding 10%. This indicates that for liquid clouds, the impact of sub-grid scale heterogeneity on the retrieval of a representative CER value is manageable and typically within an acceptable range (under 10%) for bulk microphysical properties.

**Table 2.** Retrievals from a heterogeneous cloud field with variable CDRs using POLDER-like polarized reflectances (865 nm) in 137–165° and 145–165° ranges, respectively. In all cases, the EV in the sub-scale cloud and the COT were assumed to be 0.01 and 5, respectively. The "+" indicates the equal share of the CDRs in the cloud fields. The mean CDR and EV indicate the effective radii and variances for the combined droplet size distributions. The CDR and EV estimates are restricted with $T_1 > 0.978$ and $T_2 < 0.01$.

| Combined CDRs (μm) | Sub-scale EV | Mean CDR (μm) | Mean EV | Retrievals of 137–165° | | Retrievals of 145–165° | |
|---|---|---|---|---|---|---|---|
| | | | | CDR (μm) | EV | CDR (μm) | EV |
| 5 + 10 | 0.01 | 9.00 | 0.06 | – | – | – | – |
| 5 + 15 | 0.01 | 14.00 | 0.06 | – | – | – | – |
| 5 + 20 | 0.01 | 19.12 | 0.04 | – | – | – | – |
| 10 + 15 | 0.01 | 13.46 | 0.04 | 13.0 | 0.1 | – | – |
| 10 + 20 | 0.01 | 18.00 | 0.06 | 16.5 | 0.1 | – | – |
| 15 + 20 | 0.01 | 18.20 | 0.03 | 17.5 | 0.05 | 10.0 | 0.02 |
| 5 + 10 + 15 | 0.01 | 12.70 | 0.11 | 12.0 | 0.1 | – | – |
| 5 + 10 + 20 | 0.01 | 16.92 | 0.13 | – | – | – | – |
| 5 + 15 + 20 | 0.01 | 17.35 | 0.08 | 17.5 | 0.05 | – | – |
| 10 + 15 + 20 | 0.01 | 17.07 | 0.06 | 16.0 | 0.1 | 16.5 | 0.01 |

**Figure R2.** Table 2 from Shang et al. (2015), AMT.

**Table R1.** Relative deviation between inverted CER and subpixel CER mean values.

| | Mean CER(μm) | Retrieval CER(μm) | Relative deviation |
|---|---|---|---|
| 1 | 13.46 | 13 | -3.41% |
| 2 | 18.00 | 16.5 | -8.33% |
| 3 | 18.20 | 17.5 | -3.85% |
| 4 | 12.70 | 12.0 | -5.51% |
| 5 | 17.35 | 17.5 | +0.86% |
| 6 | 17.07 | 16.0 | -6.27% |

[Figure]

**Figure R3.** Comparison of the average CER profile (averaged by height) within the same POLDER pixel versus the CER profile closest to the center of the POLDER pixel.

(c) To further investigate whether the relationship derived from CloudSat could be applied to POLDER data, we statistically analyzed the CloudSat CER profiles corresponding to these 8 POLDER pixels. For each POLDER pixel, we averaged the CloudSat CER profiles and compared them with the CloudSat profile closest to the center of the POLDER pixel. The results are shown in Fig. R3. We conclude that, except for a slightly higher deviation in Case 2, the deviations in other cases are relative small. That is, the profiles at the coarse resolution of the POLDER level (pixel_average) show little difference from the CloudSat profiles at normal resolution that we selected (center_nearest), demonstrating a high degree of similarity.

In summary, we believe that although single-layer stratiform liquid clouds exhibit spatial heterogeneity, this heterogeneity is relatively weak. This allows the relationships derived at CloudSat resolution to be applied to coarser-resolution POLDER data. However, we acknowledge that this spatial averaging inherent to coarse resolution data is the primary challenge when inferring detailed vertical profile features, as discussed in our response to the other referee's similar concern. In the future, as the observational capabilities of passive multi-angle polarization payloads improve, the association between active and passive observation data will become even stronger.

**Detailed modifications are as follows:** We have added the description of match-up process in Section 3.4. "To validate the profile structural characteristics retrieved by passive satellite observations, a match-up process between POLDER and CloudSat observations is conducted. We focus on March 2007 and identified coincident orbits that contained a high number of stratiform cloud profiles exhibiting a triangle-shaped vertical structure in CloudSat data. A specific dataset from March 2, 2007 (POLDER observation time between 06:41:09 and 07:24:06 UTC) is selected for detailed analysis in Section 4.4. The POLDER-3 Level 2 (RB2) product served as the primary dataset for matching with CloudSat observations. With a spatial resolution of approximately 16 km, this product is notably coarser than CloudSat's resolution

of less than 2 km. To establish correspondence between the datasets, the Euclidean distance between each POLDER-3 RB2 pixel center and all CloudSat data points within the POLDER-3 RB2 pixel is computed. Owing to the resolution discrepancy, a single POLDER-3 RB2 pixel often contains multiple CloudSat data points. In such cases, only the CloudSat data point closest to the center of the POLDER_RB2 pixel is retained.

Through the matching process, cloud optical thickness (COT), latitude, longitude, and other relevant data are extracted from the POLDER-3 RB2 product. These coordinates are then used to extract cloud base height (CBH), cloud top height (CTH), and cloud-top effective radius (CT_CER) obtained through the retrieval algorithm. CBH and CTH are retrieved from the POLDER-3 L1 product, which has a native resolution of 6 km, matching the resolution of the source data. CT_CER is retrieved from the POLDER L1 product at a 50 km resolution."

Meanwhile, we have added a discussion regarding the uncertainties arising from the coarse resolution of POLDER in Section 5. "The coarse resolution of POLDER products restricts the ability to capture sub-pixel cloud heterogeneity; however, by concentrating on relatively uniform single-layer stratiform liquid clouds, this study partially mitigates the resulting retrieval uncertainties. It should be noted that sub-pixel heterogeneity can inevitably introduce certain errors, particularly at cloud boundaries. Nevertheless, Shang et al. (2015) pointed out that the error caused by sub-pixel heterogeneity in cloud effective radius (CER) retrieval does not exceed 10%, which remains within an acceptable range."

**Minor Comment 1:** Line 136: Why 2013, 2019, and the first eight months of 2020? This seems like a very arbitrary group of years to use.

**Response:** The choice of data from 2013, 2019, and the first eight months of 2020 for this study was carefully considered. Our aim was to explore cloud profile structures by combining CloudSat observations with polarized multi-angle payload data. In our preliminary work, we gathered available polarized multi-angle measurements from sources such as the French POLDER-3/PARASOL instrument, as well as China's DPC/GF-5 and DPC/GF-5(02) sensors. Based on our initial assessments, CloudSat's key CWC_RO product provides reliable data between 2006 and August 2020, while POLDER-3's useful dataset covers 2005 to 2013. Additionally, we had access to China's DPC/GF-5 and DPC/GF-5(02) data, though it should be noted that these datasets are not publicly available. However, the DPC data at our disposal is limited to 2019 and 2020. To ensure our analysis remains as up-to-date as possible while still allowing for joint active-passive sensor studies, we ultimately selected CloudSat data from 2013, 2019, and the first eight months of 2020 for this investigation.

**Minor Comment 2:** Line 204: As far as I am aware, there is no "Colorado State University regional climate model." Do you mean the CSU Regional Atmospheric Modeling System (RAMS)?

**Response:** Thank you for your reminding, we feel sorry for our carelessness. In our

resubmitted manuscript, we have corrected the "Colorado State University regional climate model" to "the Colorado State University Regional Atmospheric Modeling System (RAMS)".

**Minor Comment 3:** It should be noted that the CloudSat CWC-RO product misses many (perhaps the majority of) single-layer liquid clouds, either because the clouds are masked by surface clutter or because they are below the radar's noise threshold (e.g., Lamar et al., 2020; Schulte et al., 2023). So the true nonprecipitating-to-precipitating ratio is likely much higher.

**Response:** We agree with this valuable comment. we have read the relevant papers carefully, CloudSat's data may indeed have this problem, so we try to expand the data scope to increase the amount of research data (single-layer liquid cloud). The ratio of non-precipitating clouds to precipitating clouds here is just a statistic of the data situation of our existing study, as you said, it may be different from the real ratio of non-precipitating clouds to precipitating clouds, the real ratio of non-precipitating clouds to precipitating clouds may be much higher, and we added this point to the article, as well as the possible uncertainty of CloudSat in the detection of single-layer liquid clouds.

**Minor Comment 4:** Line 240: I believe you mean Table A4 here, but even so, I do not understand what the table is intended to show.

**Response:** Yes, this refers to Table A4, which exists in order to explain the complex situation "*Other*" such profiles, there is also a part of the profile that is highly similar to the two main shapes derived from this study, and exists in order to make the shape analysis of the profiles more complete. It should be recognized that our interpretation of Table A4 is not complete, and we have added explanations in the note of Table A4: situation1 refers to a situation where only one segment of the profile does not correspond to the increasing and then decreasing shape profile of shape1, and situation2 refers to a situation where only one segment of the profile does not correspond to the monotonically decreasing shape profile of shape2. There is an intersection of situation1 and situation2, i.e., a profile that matches both situation1 and situation2 (*Intersection of 1+2*), which needs to be subtracted out when calculating the sum of the two in order to avoid double counting.

**Minor Comment 5:** Line 327: Any idea whether these two density centers have physical meaning?

**Response:** Thank you for raising this insightful question. From the Fig. 6(m), (n), and (p) of the original manuscript, i.e., the following Fig. R4(a1), (b1), and (c1), it can be observed that the scatter density distribution of the turning point CER(TP_CER) and the turning point LWC (TP_LWC) exhibits two density centers. This indicates that the relationship between TP_LWC and TP_CER is not a simple linear correlation. We conducted further analysis on the density centers, taking Fig. R4(a1) (sea non-precipitation clouds) as an example: the TP_CER shows a unimodal distribution clustered around 11–13μm, while TP_LWC exhibits a bimodal

distribution within the same TP_CER range of 11–13μm. In other words, at the same TP_CER, some profiles have relatively higher TP_LWC, while others have relatively lower TP_LWC.

[Figure]

**Figure R4.** Scatter density plots exhibiting dual density centers and their corresponding probability density distribution of TP_LWC within a specific TP_CER range.

We propose that the two density centers reflect two dominant mechanisms governing cloud microphysical processes. One mechanism is primarily dominated by condensational growth, characterized by higher liquid water content for a given cloud droplet size. This typically occurs under conditions of low cloud condensation nucleus (CCN) concentration and a stable environment, where cloud droplets grow slowly through vapor condensation and accumulate liquid water. The other mechanism is dominated by collision-coalescence growth, exhibiting lower liquid water content for the same cloud droplet size. This often happens in environments with high CCN concentrations and dynamic activity, where cloud droplets grow rapidly through collision and coalescence, leading to the redistribution of liquid water into a fewer number of larger droplets. This conclusion is strongly supported by the observed land-sea

contrast: for sea-based clouds, the density center with higher liquid water content (as seen in Figures R4(b1) and (c1)) shows a higher concentration of data points, while for continental clouds, the density center with lower liquid water content (Figure R4(a1)) is more densely populated. Over the sea, the condensation-dominated mechanism—characterized by high liquid water content—is more prevalent, consistent with the typically low aerosol concentrations, abundant moisture supply, and stable thermodynamic conditions in marine environments. In contrast, over land, the collision-coalescence-dominated mechanism—associated with lower liquid water content—prevails, aligning with the high aerosol concentrations, strong convective activity, and dynamically active nature of continental settings. This systematic geographical pattern strongly affirms the physical reality of the dual density centers, demonstrating that they represent distinct cloud microphysical states driven by environmental factors such as aerosol concentration and thermodynamic conditions.

**Detailed modifications are as follows:** We have briefly expanded on the potential physical implications of the dual density centers in the original manuscript. "The two density centers observed in the relationship between the TP_CER and TP_LWC reflect two distinct cloud microphysical regimes. One is primarily driven by condensational growth, which tends to occur under low aerosol and stable conditions, resulting in higher LWC for a given droplet size. The other is dominated by collision-coalescence, typical in relative high aerosol and dynamically active environments, leading to lower LWC for the same droplet size."

**References:**

Jr., R. A. H.: Cloud Dynamics, 2nd, Academic Press 2014.

Shang, H., Chen, L., Bréon, F. M., Letu, H., Li, S., Wang, Z., and Su, L.: Impact of cloud horizontal inhomogeneity and directional sampling on the retrieval of cloud droplet size by the POLDER instrument, Atmos. Meas. Tech., 8, 4931-4945, 10.5194/amt-8-4931-2015, 2015.

Shang, H., Letu, H., Bréon, F.-M., Riedi, J., Ma, R., Wang, Z., Nakajima, T. Y., Wang, Z., and Chen, L.: An improved algorithm of cloud droplet size distribution from POLDER polarized measurements, Remote Sensing of Environment, 228, 61-74, https://doi.org/10.1016/j.rse.2019.04.013, 2019.

---

## Author Comment (AC3)

Dear referees,

Thank you for your comments concerning our manuscript entitled *"Characterization of liquid cloud profiles using global collocated active radar and passive polarimetric cloud measurements" (ID: egusphere-2025-2471)*. Those comments are all valuable and very helpful for revising and improving our paper, as well as the important guiding significance to our researches. We have studied comments carefully and have made correction which we hope meet with approval. Revised portions are marked in blue (referee #1) /orange (referee #2) in our response document. The relevant references are at the end of our reply letter. The main corrections in the paper and the responses to the referees' comments are as following:

**Response to Referee #2's Comments**

**Overall Comment:**

This study presents a significant methodological advancement in remote sensing of liquid cloud vertical profiles by integrating active (CloudSat radar) and passive (POLDER polarimetric imager) measurements. The work is rigorous and innovative, leveraging global-scale data (12.47M profiles) to classify cloud profile shapes, establish statistical correlations, and develop retrieval models. The paper does several things exceptionally well: 1. Global-scale profile classification is statistically robust, providing a comprehensive understanding of dominant cloud structures (triangle-shaped and monotonically decreasing profiles). 2. Shape simplification using the VM algorithm is clever, effectively reducing noise while preserving key structural features. 3. Correlation analyses (e.g., between TP_CER and CB_CER, LWP) provide valuable physical insights into cloud microphysics. 4. The hybrid ML/physical modeling approach (CPRM reconstruction) is methodologically sound, combining the strengths of data-driven and physics-based methods.

However, passive retrieval limitations and dependence on CloudSat for prior knowledge (especially CB_CER estimation) remain fundamental constraints. Current methods cannot fully resolve vertical uncertainties due to: 1. CloudSat's 240-m vertical resolution, which restricts layer precision (>2.64 km); 2. POLDER's coarse spatial resolution (50 km for CER), which may misrepresent fine-scale cloud variations.

Despite these limitations, this work represents a significant step toward parameterizing cloud processes in climate models , particularly for stratiform/stratocumulus clouds.

**Major Comment 1:** The study's heavy reliance on CloudSat-derived empirical relationships for estimating CB_CER represents a critical constraint that significantly impacts the broader applicability of the methodology. While the multivariate regression approach using CT_CER, LWP, and CTH demonstrates reasonable correlation (0.75-0.92), this dependence on active sensor data fundamentally undermines the potential for truly independent passive retrievals. The propagation of errors through this empirical relationship is particularly concerning, with RMSE values reaching up to 1.96 μm for land-based clouds. This limitation is especially problematic because CB_CER serves as a foundational parameter for estimating TP_CER, meaning any errors in the initial cloud-base estimation will cascade through the entire retrieval process. The authors should more thoroughly discuss potential mitigation strategies, such as incorporating ancillary data sources or developing physics-based approaches to reduce this critical dependency on CloudSat for prior knowledge.

**Response:** We sincerely thank your thoughtful consideration of the empirical relationship derived from CloudSat data used to estimate CB_CER in this study. We fully acknowledge the limitations of this approach. Currently, directly retrieving the cloud-bottom effective particle radius using passive observational data involves significant uncertainty and is considerably challenging to achieve. Few studies have attempted to retrieve the cloud-bottom effective radius. In Platnick (2000), the relative contribution (vertical weighting) of different cloud layers to the overall retrieval was simulated using the adding-doubling method. The results indicated that the cloud base contributes the least to the total reflectance, with its contribution approaching zero for optically thick clouds. Additionally, Buggee and Pilewskie (2025) noted that constraining cloud-bottom droplet size is highly difficult, as the average penetration depth of visible and near-infrared reflectance is limited to the vicinity of the cloud top. It is challenging to retrieve the cloud-bottom effective particle radius using passive observational data based on classical physical retrieval methods (radiative transfer model simulations and look-up tables), due to the low sensitivity of visible and infrared bands currently used for cloud detection to cloud-bottom signals.

However, if cloud microphysical profiles are to be obtained from passive satellite measurements, cloud-bottom droplet size remains an unavoidable yet challenging issue. This study presents a preliminary attempt to estimate the cloud-bottom effective radius for single-layer stratiform liquid clouds using an empirical relationship derived from CloudSat data. All input variables are based solely on features obtainable from passive observations. The RMSE for the TP_CER retrieval of sea non-precipitation clouds, sea precipitation clouds, land non-precipitation clouds, and land precipitation clouds reached 1.19, 1.30, 1.76, and 1.96μm, respectively. The retrieval accuracy of TP_CER for sea-based clouds is higher than that for land-based clouds, while the accuracy for non-precipitation clouds is superior to that for precipitation clouds. We consider the lowest accuracy in estimating the TP_CER for land precipitation clouds to be a reasonable result, attributable to the following factors: (1) The ocean surface exhibits simple, homogeneous, and highly uniform properties, whereas land surfaces are characterized by complex types, low heat capacity, pronounced diurnal and spatial variations in surface temperature, and higher aerosol concentrations that provide more abundant cloud condensation nuclei. (2) Non-precipitation clouds are relatively homogeneous, with thinner layers and limited vertical development, whereas precipitation clouds exhibit more intense microphysical processes. Current studies on CER retrieval methods generally report RMSE values ranging between 4–7μm (Ma and Husi, 2024; Du et al., 2024). This indicates that significant uncertainty remains in CER retrieval, and our TP_CER estimation scheme achieves relatively favorable accuracy in comparison.

Certainly, we fully understand the referees' concerns regarding this estimation approach: (1) The empirical relationship is derived entirely from active sensor observations, specifically CloudSat CPR data; (2) The retrieval error in CB_CER, which serves as the initial parameter for estimating TP_CER, may lead to an accumulation of errors in the TP_CER estimation. We also have plans to further improve this preliminary approach in subsequent research. In response to the referee' feedback, we have added a discussion in the final chapter of the paper

outlining potential strategies to mitigate these uncertainties: (1) Introduce meteorological factors—such as ERA5 reanalysis data (e.g., wind speed, temperature, humidity, pressure, vertical velocity)—to assist in estimating the cloud-bottom effective radius, thereby enhancing the physical characterization of the cloud-base environment; (2) Optimize existing methods for directly retrieving cloud-bottom particle size using passive observations (Level 1 or Level 2 products), improve the robustness of the retrieval model, and clarify the applicable boundaries of the method—specifically, determining the optical thickness threshold beyond which passive observations can no longer capture cloud-bottom information; (3) Incorporate an uncertainty weighting framework to dynamically adjust the contribution weights of different input parameters based on their reliability, thereby refining the retrieval accuracy of CB_CER and reducing dependence on CloudSat-derived empirical relationships.

**Detailed modifications are as follows:** In Section 5 of the manuscript, we clarified that the current uncertainties in cloud base CER retrieval remain a major source of error in the vertical structure retrieval of cloud profiles. Additionally, potential improvement strategies are discussed based on your suggestions. "In terms of estimating the CB_CER: (1) Introduce meteorological factors—such as ERA5 reanalysis data (e.g., wind speed, temperature, humidity, pressure, vertical velocity)—to assist in estimating the CB_CER, thereby enhancing the physical characterization of the cloud-base environment; (2) Optimize existing methods for directly retrieving cloud-bottom particle size using passive observations (Level 1 or Level 2 products), improve the robustness of the retrieval model, and clarify the applicable boundaries of the method—specifically, determining the optical thickness threshold beyond which passive observations can no longer capture cloud-bottom information; (3) Incorporate an uncertainty weighting framework to dynamically adjust the contribution weights of different input parameters based on their reliability, thereby refining the retrieval accuracy of CB_CER and reducing dependence on CloudSat-derived empirical relationships."

**Major Comment 2:** The observed inconsistencies in POLDER-based TP_CER and TP_NCOT retrievals, particularly evident in Cases 1, 2, and 7 of Figure 9, reveal important limitations in current passive sensing capabilities. These discrepancies likely stem from multiple compounding factors that warrant deeper examination. First, POLDER's relatively coarse spatial resolution (approximately 50 km for CER retrievals) means it cannot resolve fine-scale heterogeneities in cloud microphysical properties that CloudSat can detect along its narrow swath. Second, the visible-band measurements used by POLDER are primarily sensitive to cloudtop properties, making it inherently challenging to accurately characterize vertical microphysical gradients. The study would benefit from a more detailed error analysis quantifying how these sensor limitations translate to uncertainties in profile reconstruction, perhaps through sensitivity studies or comparison with higher-resolution datasets where available.

**Response:** We sincerely thank the referee for this insightful observation and for highlighting this crucial aspect of our study. The retrieval of the TP_CER and its location in Cases 1, 2, and 7 in Figure 9 exhibits slightly higher errors compared to other cases. We fully agree that a more detailed error analysis is crucial to thoroughly investigate the causes and better understand the

limitations of passive sensor data in cloud profile retrieval. The main sources of uncertainty are considered to be the following:

(1) The coarse resolution of POLDER products cannot capture inherent subpixel heterogeneity. This study utilized the POLDER Level 2 RB2 product (16 km resolution) along with cloud top height (CTH), cloud base height (CBH) (both at 6 km), and cloud top effective radius (CER) data (50 km resolution), all of which were derived from POLDER Level 1 products. Compared to CloudSat data, the coarser resolution of POLDER may cause biases resulting from subpixel heterogeneity. We conducted a further analysis of the eight cases in Fig. 9 by averaging the CloudSat CER profiles within each corresponding POLDER pixel along the altitude dimension. This process effectively aggregates the high-resolution CloudSat profiles to the spatial scale of a POLDER pixel, simulating what POLDER would likely "see". The resulting averaged profiles were then compared against our validation data—the CloudSat profile closest to the center of the POLDER pixel—as shown in the figure below. Although this study specifically targets horizontally relatively homogeneous single-layer stratiform water clouds, subpixel heterogeneity—resulting from POLDER's coarse resolution—remains one of the main sources of error in estimating the structural parameters of cloud profiles.

[Figure]

**Figure R1.** (It appeared in the previous text, so it has been repeatedly numbered here.) Comparison of the average CER profile (averaged by height) within the same POLDER pixel versus the CER profile closest to the center of the POLDER pixel.

(2) Inherent physical limitation: Vertically integrated signal. The retrieval of CER from POLDER observations differs from traditional dual-channel methods, relying instead on the directional characteristics of polarized reflectance within the cloud bow scattering angle range. It should be emphasized that all such passive retrieval techniques essentially provide a vertically integrated measurement—a weighted average signal sensitive to microphysical properties from the cloud top downward through a depth determined by cloud optical thickness. This fundamental characteristic inherently increases the uncertainty in retrieving vertical structural features. This limitation is inherent not only to POLDER but to all passive sensor payloads, which underscores the significance of our study's exploration of cloud profile retrieval using passive data.

(3) Propagation of input uncertainties. Errors in the upstream inputs (COT, CT_CER from POLDER, and CTH/CBH from the combination of POLDER and ancillary data) inevitably propagate into the LWP and CB_CER. This accumulated uncertainty can then propagate into errors in the final estimated profile parameters, including the TP_CER and its location. Cases with higher sub-pixel heterogeneity or where the cloud-top CER is less representative of the layer-average are particularly susceptible to this propagation effect.

Certainly, we believe these errors can be mitigated through the ongoing development of polarized multi-angle sensors. A number of spaceborne instruments with higher spatial resolution, additional observational channels, and expanded viewing angles have recently been launched or are planned for future missions. Examples include the already deployed DPC/GF-5, which achieves a nadir resolution of 3.3 km, and the upcoming 3MI/Metop-SG sensor developed by the European Space Agency, which offers a nadir resolution of 4 km along with enhanced scalar and polarimetric observations in the near-infrared bands. These advances provide promising opportunities for improving the accuracy of cloud profile retrieval.

**Detailed modifications are as follows:** We have revised and expanded the explanation regarding the causes of and error analysis for the relatively larger errors in Cases 1, 2, and 7 compared to other cases in Section 4.3. "The coarse resolution of POLDER products cannot capture inherent subpixel heterogeneity. This study utilizes the POLDER Level 2 RB2 product (16 km resolution) along with cloud top height (CTH), cloud base height (CBH) (both at 6 km), and cloud top effective radius (CER) data (50 km resolution), all of which are derived from POLDER Level 1 products. Compared to CloudSat data, the coarser resolution of POLDER may cause biases resulting from subpixel heterogeneity. We conduct a further analysis of the eight cases in Fig.A2 by averaging the CloudSat CER profiles within each corresponding POLDER pixel along the altitude dimension. This process effectively aggregates the high-resolution CloudSat profiles to the spatial scale of a POLDER pixel, simulating what POLDER would likely "see". The resulting averaged profiles are then compared against our validation data—the CloudSat profile closest to the center of the POLDER pixel. Although this study specifically targets horizontally relatively homogeneous single-layer stratiform water clouds, subpixel heterogeneity—resulting from POLDER's coarse resolution—remains one of the main sources of error in estimating the structural parameters of cloud profiles. (2) Inherent physical limitation: Vertically integrated signal. The retrieval of CER from POLDER observations differs from traditional dual-channel methods, relying instead on the directional characteristics of polarized reflectance within the cloud bow scattering angle range. It should be emphasized that all such passive retrieval techniques essentially provide a vertically integrated measurement—a weighted average signal sensitive to microphysical properties from the cloud top downward through a depth determined by cloud optical thickness. This fundamental characteristic inherently increases the uncertainty in retrieving vertical structural features. (3) Propagation of input uncertainties. Errors in the upstream inputs (COT, CT_CER from POLDER, and CTH/CBH from the combination of POLDER and ancillary data) inevitably propagate into the LWP and CB_CER. This accumulated uncertainty then propagates into errors in the final estimated profile parameters, including the TP_CER and its location. Cases with higher sub-pixel heterogeneity or where the cloud-top CER is less

representative of the layer-average are particularly susceptible to this propagation effect." Meanwhile, we have included Figure R1 in the appendix (Figure A2).

**Major Comment 3:** CloudSat's 240-m vertical resolution, while impressive for spaceborne radar, imposes significant limitations on the study's ability to characterize thin or finely structured cloud layers. This resolution threshold means the CPRM reconstruction cannot resolve features smaller than about 2.64 km in vertical extent, potentially missing important microphysical transitions in shallow cloud systems. The impact is particularly relevant for stratocumulus clouds, which often exhibit thin but meteorologically important structures like sharp inversion layers or thin drizzling layers near cloud base. The authors should expand their discussion of how this resolution limitation affects the physical interpretation of their results, and perhaps suggest how future sensors with finer vertical resolution (like EarthCARE's radar) might overcome this constraint.

**Response:** We fully agree with the referees' perspective that the 240m vertical resolution of CloudSat is indeed insufficient to capture the fine structure of thin cloud layers. Here, "fine structure" may have two implications: 1) cloud layers with a thickness less than 240m; 2) cloud layers with a thickness greater than 240m, but with local fine structures within the cloud that are less than 240m, which the radar cannot detect, such as sharp inversion layers or thin drizzling layers near the cloud base. Therefore, in this study, we avoided selecting overly thin cloud layers when choosing profiles, using a minimum of three bins as a primary criterion for profile data screening. There are two main reasons for this: 1) overly thin cloud layers cannot be represented structurally by CloudSat, such as shape characteristics; 2) even if CloudSat can record overly thin cloud layers, it cannot capture some fine features at the cloud top or cloud base. We believe that the formation conditions for layered clouds are similar, so thicker layered clouds should have a structure similar to thin clouds and amplify local fine structures. Therefore, this study primarily focuses on profiles with a bin count of 3 or more, which may overlook some thin clouds but will not affect the overall statistical analysis results. Additionally, we would like to clarify that our unclear wording may have caused your misunderstanding. Line 266 mentions that the primary research object of this paper is CloudSat cloud profiles with vertical thicknesses less than 2.64 km. The intended meaning is that our statistical analysis found that the majority of single-layer stratiform liquid cloud profiles observed by CloudSat have bin counts less than 11 (each bin representing a range of 240 m). Therefore, the study focuses on profiles with vertical thicknesses of 2.64 km or less. All statistical analyses of profiles in this paper are based on data from profiles with vertical thicknesses of 2.64 km or less. This does not imply that CPRM cannot reconstruct profiles smaller than 2.64 km; in fact, CPRM can reconstruct profiles of arbitrary thickness, which primarily depends on the input data. (Shang et al., 2025).

Meanwhile, based on your feedback, we conducted a survey of the instrument characteristics of EarthCARE CPR and compared them with those of CloudSat CPR (Table R1). EarthCARE CPR primarily offers the following technical advantages: 1) Higher vertical solution: EarthCARE/CPR has a vertical sampling resolution of 100m, representing a significant improvement over CloudSat CPR's 240m resolution; 2) Higher sensitivity: EarthCARE/CPR's sensitivity has been improved to approximately -35 dBZ, enabling more precise detection of

cloud structure; 3) Doppler wind measurement capability: Compared to CloudSat, EarthCARE's CPR has Doppler capability, enabling the first-ever measurement of vertical wind speeds within clouds by a satellite-borne radar (with an accuracy of 0.5m/s); 4) Synchronized observation capability: EarthCARE CPR can synchronize observations with instruments like ATLID, offering a significant advantage over the loose coordination between CloudSat and CALISPO. However, due to insufficient data accumulation for EarthCARE at present, CloudSat data still holds high research value.

**Table R1:** Comparison of Instrument Characteristics of EarthCARE CPR and CloudSat CPR

| Instrument Characteristics | CloudSat CPR | EarthCARE CPR |
|---|---|---|
| Mission duration | 2006-2020 | 2024- |
| Frequency | 94.05 GHz | 94.05 GHz |
| Altitude | 705 km | 400 km |
| Hor. res. at nadir | 1.2 km | 500 m |
| Vert. resolution | 500 m (240m sampling) | 500 m (100m sampling) |
| Doppler capability | N/A | Yes |
| Sensitivity | -28 dBZ | -35 dBZ |

**Detailed modifications are as follows:**

(1) We have added a discussion on the resolution limitations of CloudSat in the uncertainty analysis of Section 5. "The 240-m vertical resolution of CloudSat is indeed insufficient to resolve ultra-thin cloud layers or capture fine-scale intra-cloud structures, such as sharp inversion layers or thin drizzling layers near cloud base."

(2) The potentially misleading expression at the end of the first paragraph in Section 4.1 of the manuscript has been revised. "Based on the above analysis, given CloudSat's vertical resolution of 240 m per bin and the fact that approximately 99.5% of the profiles are concentrated within 11 bins, this study focuses on single-layer stratiform liquid clouds with a geometric thickness of less than 2.64 km."

(3) Related suggestion has been added in Section 5 of the paper regarding how future sensors with finer vertical resolution (such as EarthCARE's radar) could potentially overcome this limitation. "For the issue that CloudSat is insufficient to distinguish thin clouds or fine cloud structures smaller than 240m, in the future, the EarthCARE/CPR observation data with higher vertical resolution can be adopted to alleviate this problem. EarthCARE/CPR demonstrates notable advancements over CloudSat, most significantly through its finer vertical resolution of 100 m compared to CloudSat's 240 m. Additional enhancements include higher detection sensitivity, Doppler-based vertical wind measurements, and synchronized multi-sensor observational capabilities. These improvements are expected to deliver enhanced observational capabilities for characterizing finer-scale cloud microphysical processes and their interactions with atmospheric dynamics."

**Major Comment 4:** The coarse resolution of POLDER observations presents multiple challenges that extend beyond the immediate retrieval accuracy issues. At 50 km resolution, individual POLDER pixels often or possibly integrate across multiple cloud regimes, potentially blending

fundamentally different cloud types and obscuring important spatial gradients. This becomes particularly problematic when trying to apply the methodology to broken cloud fields or cloud edges, where sub-pixel variability is high. While the current focus on stratiform clouds is understandable given their relative homogeneity, the paper would benefit from a more thorough discussion of how partial cloudiness and three-dimensional radiative effects might bias the retrievals. The authors might consider adding a sensitivity analysis or at least a more detailed qualitative discussion of these effects in the limitations section.

**Response:** We sincerely thank the referee for raising this important issue regarding the impact of sub-pixel cloud heterogeneity on retrieval accuracy, which is indeed an important challenge associated with the coarse spatial resolution of POLDER observations. The concern that single pixels may integrate multiple cloud regimes—particularly in broken cloud fields or near cloud edges—is well taken, and we agree that such blending could obscure spatial gradients and introduce potential biases in the retrieved parameters.

In response to this comment, we will expand the "Limitations" section (in Section 5) to include a more thorough discussion of how partial cloudiness and three-dimensional radiative effects may influence our results. Specifically, we will reference our previous quantitative analysis (Shang et al., 2015), which systematically evaluated the effects of sub-pixel cloud heterogeneity on POLDER cloud microphysical retrievals. That study provides concrete uncertainty estimates that help contextualize the potential biases in the current application. We will also clarify that, although this study focuses on relatively homogeneous stratiform clouds to minimize such effects and assess methodological feasibility, the presence of unresolved cloud variability near cloud edges remains an important source of uncertainty.

Additionally, we will emphasize that future satellite missions with higher spatial resolution—such as the Directional Polarimetric Camera (DPC) and the Multi-viewing Multi-channel Multi-polarization Imager (3MI)—will help alleviate these challenges by reducing sub-pixel variability. This expanded discussion will provide readers with a clearer understanding of the current limitations and pathways for future improvement.

Thank you again for highlighting this critical aspect of our work.

**Detailed modifications are as follows:** We have incorporated a related discussion in the limitations section (Section 5). "The primary challenges in retrieving profile structural features originate from the following aspects: (1) The coarse resolution of POLDER products restricts the ability to capture sub-pixel cloud heterogeneity; however, by concentrating on relatively uniform single-layer stratiform liquid clouds, this study partially mitigates the resulting retrieval uncertainties. It should be noted that sub-pixel heterogeneity can inevitably introduce certain errors, particularly at cloud boundaries. Nevertheless, Shang et al. (2015) pointed out that the error caused by sub-pixel heterogeneity in cloud effective radius (CER) retrieval does not exceed 10%, which remains within an acceptable range." A feasible improvement plan for future work has also been proposed. "To address the issues mentioned above, the improvement strategies below can be implemented: (1) Internationally, there are currently polarimetric multiangle payloads with higher spatial resolution and greater observation angles that have been launched or are planned for launch. For instance, China's DPC/GF-5 achieves a spatial resolution of nadir 3.3 km; the 3MI/Metop-SG developed by the European Space Agency offers a spatial resolution of nadir 4 km, supports up to 21 observation angles, and incorporates near-infrared bands. These capabilities collectively enable higher-resolution CER retrieval."

**Major Comment 5:** The exclusion of 9.1% of profiles classified as complex-shaped introduces a subtle but potentially important selection bias in the results. While this filtering improves the clarity of the statistical relationships, it risks creating an overly idealized representation of real-world cloud profiles. Many meteorologically significant situations - such as clouds undergoing strong entrainment, multilayered structures, or precipitating systems - may fall into this excluded category. The authors should more thoroughly justify their exclusion criteria and discuss how this might affect the generalizability of their findings. A sensitivity analysis showing how including some portion of these complex profiles affects the retrieval statistics would significantly strengthen the paper's conclusions.

**Response:** We sincerely appreciate the referee's insightful comment regarding the potential selection bias introduced by excluding complex-shaped cloud profiles (9.1% of the dataset). We agree that this is an important consideration, and we have taken the following steps to address this concern in the revised manuscript:

1) Justification of Exclusion Criteria

We have supplemented the criteria for determining complex shape profiles, which are defined as profiles that do not belong to any of the following shapes: (1) triangle shaped (Inc_Dec), increasing then decreasing; (2) monotonically decreasing (Mono_Dec); (3) monotonically increasing (Mono_Inc); and (4) decreasing then increasing (Dec_Inc). These shapes depict the vertical variation in the CER from the cloud base to the top. The four shapes can be simply expressed by the following formulas:

$$\text{Inc\_Dec: } CER_1 < CER_2 < \cdots < CER_k > CER_{k+1} > \cdots > CER_N, 1 < k < N$$
$$\text{Mono\_Dec: } CER_1 > CER_2 > \cdots > CER_N$$
$$\text{Mono\_Inc: } CER_1 < CER_2 < \cdots < CER_N$$
$$\text{Dec\_Inc: } CER_1 > CER_2 > \cdots > CER_k < CER_{k+1} < \cdots < CER_N, 1 < k < N$$

Where $CER_i$ denote the CER at the $i$-th vertical level (bin), $i$=1 corresponds to the cloud base and $i$=N to the cloud top.

2)Further analysis and discussion on the 9.1% complex-shaped profile

This paper further analyzes the similarity between the 9.1% complex-shaped profile and the two main shapes extracted in this study (first increasing then decreasing, monotonically decreasing) in Table A4. The purpose of this table is to make the shape analysis more complete. In table A4, situation1 refers to a situation where only one segment of the profile does not correspond to the increasing and then decreasing shape profile of shape1, and situation2 refers to a situation where only one segment of the profile does not correspond to the monotonically decreasing shape profile of shape2. There is an intersection of situation1 and situation2, i.e., a profile that matches both situation1 and situation2 (*Intersection of 1+2*), which needs to be subtracted out when calculating the sum of the two in order to avoid double counting. Table 4

shows that among the 9.1% of complex-shaped profiles, approximately 60% of the profiles differ from the main profile shape we extracted in only one segment, which also proves that the main profile shape we extracted is robust and universal. Although our profile simplification program can reduce complex shapes to simpler forms, there is controversy regarding the specific categories to which these shapes belong. Taking Fig. R2 as an example, Complex Shape 1 and Complex Shape 2 can be simplified into different primary shapes. Therefore, we believe that these 9.1% of complex profile shapes can be further analyzed in subsequent studies, but it is unnecessary to include them in the follow-up retrieval research presented in this paper, as their inclusion would introduce unnecessary errors into our training process.

[Figure]

**Figure R2.** Complex CER profile shapes and their possible corresponding simplified shapes (examples)

**Detailed modifications are as follows:** Based on your suggestions, we have made the following modifications to the second paragraph of Section 4.1 (Typical shape and structural characteristic analysis of CER profiles) in the original manuscript. "Through an extensive literature review and visual analysis of CloudSat single-layer liquid cloud profiles, the vertical variation of cloud effective radius (CER) can be classified into four distinct shapes based on the monotonicity between adjacent layers: (1) triangle shaped (Inc_Dec), increasing then decreasing; (2) monotonically decreasing (Mono_Dec); (3) monotonically increasing (Mono_Inc); and (4) decreasing then increasing (Dec_Inc). These shapes depict the vertical variation in the CER from the cloud base to the top. The four shapes can be simply expressed by the following formulas:

$$\text{Inc\_Dec: } CER_1 < CER_2 < \cdots < CER_k > CER_{k+1} > \cdots > CER_N, 1 < k < N \quad (3)$$

$$\text{Mono\_Dec: } CER_1 > CER_2 > \cdots > CER_N \quad (4)$$

$$\text{Mono\_Inc: } CER_1 < CER_2 < \cdots < CER_N \quad (5)$$

$$\text{Dec\_Inc: } CER_1 > CER_2 > \cdots > CER_k < CER_{k+1} < \cdots < CER_N, 1 < k < N \quad (6)$$

Where $CER_i$ denote the CER at the $i$-th vertical level (bin), $i$=1 corresponds to the cloud base and $i$=N to the cloud top. Systematic classification and statistical analysis confirm these patterns (Fig. 3(a)). Collectively, these four shapes account for 90.1% of the observed CER profiles, with Shapes

2 (Mono_Dec: 48.8%) and 1 (Inc_Dec: 39.7%) being the most prevalent, highlighting their dominance in the liquid stratiform cloud life cycle. The remaining 9.1% represent complex-shaped profiles that do not conform to these four categories, with further analysis of these cases presented in the Appendix A (Table A4). Approximately 60% of these profiles contain only single segments inconsistent with Shapes 1 and 2. Although our profile simplification program can reduce complex shapes to simpler forms, there is controversy regarding the specific categories to which these shapes belong. Taking Fig. A1 as an example, Complex Shape 1 and Complex Shape 2 can be simplified into different primary shapes. Therefore, we believe that these 9.1% of complex profile shapes can be further analyzed in subsequent studies, but it is unnecessary to include them in the follow-up parts presented in this study, as their inclusion would introduce unnecessary errors into our retrieval prior knowledge." Fig. A1 corresponds to Fig. R2 in this response.

**Major Comment 6:** The minimal three-year overlap between CloudSat (2006-2020) and POLDER (2004-2013) operations raises important questions about the dataset's representativeness for global climatological studies. Cloud properties exhibit significant interannual variability influenced by large-scale modes like ENSO, and a three-year period may not adequately capture this natural variability. The authors should discuss whether their training dataset (2013, 2019, and part of 2020) is truly representative of global cloud conditions, and whether the limited temporal sampling might introduce biases in the derived statistical relationships. This is particularly relevant given that some of the training years (like 2019) were characterized by unusual atmospheric conditions in certain regions.

**Response:** Thank you for your question. We understand your concerns regarding data representativeness. The primary focus of this study is on liquid stratiform clouds (stratus + stratocumulus) on a global scale, which has a relatively small vertical thickness and is commonly found in stable atmospheric conditions. It is unlikely to persist under extremely unstable meteorological conditions. For example, while El Niño may suppress the formation of stratiform clouds, it is unlikely to affect our statistical results. Therefore, we believe that the abnormal atmospheric conditions in certain regions in 2019 will not impact the overall liquid stratiform clouds profile dataset. Secondly, although the CloudSat profile data used in this paper spans only the nearly three years, the overall data volume is substantial (as shown in Table A3), reaching nearly 12.48 million entries, and it covers the entire globe spatially. The large data volume of this study is sufficient to ensure the robustness of the profile statistical results, and that abnormal atmospheric conditions in certain regions will not affect the overall statistical results. Since our main objective is to investigate the structural characteristics of the global stratiform liquid cloud profile rather than the interannual variability of global clouds, we consider the data from the nearly three years to be sufficiently comprehensive. Additionally, the time range of the research data selected in this paper (2013, 2019, and 2020) was not chosen arbitrarily. It was selected to match the time range of the polarized multi-angle satellite (POLDER-3/Parasol, DPC/GF-5, DPC/GF-5(02)) data. Although the overlap with the years observed by currently launched payloads and CloudSat is relatively short, there are ongoing international plans to launch W-band cloud radars (e.g., EarthCARE/CPR) and polarimetric multi-angle satellites (e.g., 3MI), making the selection of active radar and polarimetric multiangle payload data for joint cloud profile research still of significant long-term value.

**Detailed modifications are as follows:** Based on your suggestion, we discussed and explained this issue in Section 3.1. "Furthermore, since our study focuses on relatively homogeneous and stable single-layer stratiform liquid clouds, localized atmospheric anomalies do not impact the statistical results presented herein."

**Major Comment 7:** The notably weak correlations for TP_NCOT (Figure 5b) reveal fundamental challenges in passively retrieving information about vertical structure inflection points. This poor correlation performance suggests that current passive observables may lack the necessary information content to reliably determine the normalized optical thickness at turning points. The authors should expand their discussion of potential physical reasons for this limitation, such as: The insensitivity of passive measurements to vertical redistribution of cloud water; The degeneracy between different vertical configurations that produce similar top-of-atmosphere signals; The potential for cloud inhomogeneity effects to obscure the true profile characteristics. A more thorough exploration of these physical limitations would help readers better understand the boundaries of what can realistically be achieved with passive profile retrievals.

**Response:** Thank you very much for your insightful and constructive comments. You have accurately identified a critical and worthy issue in our research—the weak correlation between TP_NCOT and other cloud parameters, and the fundamental challenges it reveals in passively remote-sensing the vertical structure of clouds. First, it is important to clarify that both the TP_NCOT (i.e., the turning point location) and the related cloud parameters used in the correlation analysis in Fig. 5(b) are derived exclusively from active observations. Therefore, we believe that this weak correlation is fundamentally unrelated to the insufficient sensitivity of passive observations to vertical redistribution of cloud water or the similarity of atmospheric top signals across different vertical configurations. In other words, while these limitations do exist, they are not the cause of the phenomenon observed in Fig. 5(b). Additionally, the masking of true profile characteristics by cloud inhomogeneity effects does not occur in active data. This issue primarily stems from the limitations of passive observations, as current active data, despite some uncertainty, can more accurately reflect the true state of cloud profiles compared to passive observation data.

The essence of this phenomenon may be that the turning point position is almost unaffected by droplet size, cloud water content, cloud thickness, and other factors, and the turning point position of the cloud effective radius profile is not fundamentally related to these parameters. According to the analysis in Section 4.1, the formation of cloud profile with a turning point (first increasing then decreasing in shape) is primarily attributed to two possible causes: 1) The entrainment of dry air at the cloud top enhances droplet evaporation, leading to a reduction in droplet size and the appearance of a turning point in the cloud profile; 2) When collision and coalescence at the cloud base cause droplet size to reach a critical value, drizzle or precipitation forms at the cloud base, and the falling drizzle or precipitation may cause a reduction in droplet size at the cloud base. 1) is the primary cause of the turning point in the non-precipitation profile, while 1) and 2) may jointly contribute to the appearance of the turning point in the

precipitation profile. From the perspective of the causes of the turning point, the distribution of the normalized optical thickness at the turning point may be relatively random, potentially occurring at any position within the cloud. Fig. 3(b) also corroborates this point. Due to the vertical resolution limitations of active satellites, the turning point locations exhibit a multi-peak distribution; otherwise, they might be distributed more dispersedly. This also indirectly reflects the difficulty of estimating the TP_NCOT using these parameters through relatively conventional methods, and the even greater difficulty of retrieving the profile TP_NCOT through "path-integrated" observations using passive satellites. We have added a discussion of the possible causes of this weak correlation in Section 4.2 of the article. Thank you once again for your valuable suggestions.

**Detailed modifications are as follows:** We supplemented the relevant discussion in Section 4.2. "The weak correlation for TP_NCOT stems from the fact that the TP position is largely independent of common cloud parameters such as droplet size, cloud water content, and cloud thickness. Instead, it is primarily influenced by microphysical processes like cloud-top entrainment and precipitation formation, leading to a relatively random distribution of TP_NCOT within the cloud layer. This inherent randomness makes it inherently difficult to estimate TP_NCOT using conventional correlation-based method."

**Major Comment 8:** The substantially higher errors over land (RMSE 1.96μm vs. ~1.3μm over ocean) point to important unresolved challenges in land cloud retrievals that deserve more detailed discussion. Several factors likely contribute to this performance gap:
- Greater sub-pixel heterogeneity over land due to surface variability
- Higher and more variable aerosol loading affecting the cloud microphysics
- Stronger surface heating effects on cloud boundary layer dynamics
- Potential artifacts from the underlying terrain elevation and albedo

The paper would benefit from a dedicated discussion of these land-specific challenges and potential strategies to mitigate them, such as incorporating land surface type classifications or aerosol information into the retrieval framework.

**Response:** Thank you for your valuable feedback on our manuscript. The issue you raised regarding the significantly higher retrieval errors for key structural features of cloud profiles over land compared to oceanic regions is indeed crucial. We fully agree that this discrepancy reflects the greater challenges inherent in cloud retrieval over land. Below is a point-by-point response and additional discussion addressing your suggestions:

1. Analysis of Error Sources Over Land Regions

a) Sub-pixel Surface Heterogeneity: Variations in surface reflectance among different land cover types (e.g., vegetation, bare soil, urban areas) lead to mixed-pixel effects, complicating the decoupling of cloud optical properties.

b) Aerosol Interference: Higher and spatiotemporally variable aerosol loadings over land can perturb cloud signals either indirectly by altering cloud microphysics (e.g., through cloud condensation nuclei effects) or directly via scattering.

c) Surface Heating Effects: The lower thermal inertia of land surfaces results in more complex

boundary-layer dynamics, increasing spatiotemporal variability in cloud base height and cloud layer thickness, which in turn elevates retrieval uncertainty.

d) Interference from complex terrain and high-albedo surfaces: Complex terrain (e.g., mountains) and high-albedo surfaces (e.g., snow cover) are prone to causing false positives in cloud detection or overestimation of optical thickness.

2. Potential Improvement Strategies

In response to your suggestions, we have supplemented our discussion with potential solutions to address the significant retrieval errors in key structural characteristics of cloud profiles over land regions, including: a) Integration of land cover classification data: Incorporating higher-resolution land cover data (e.g., MODIS Land Cover products, Sentinel-2 10-meter resolution data) could help mitigate mixed-pixel effects. A zone-based retrieval strategy may be developed by establishing customized radiative transfer model parameters for distinct surface types (e.g., forest, cropland, urban areas). Seasonal variations in vegetation indices (e.g., NDVI) could also be employed to dynamically adjust surface albedo parameters. b) Integration of aerosol ancillary data: Multi-source aerosol observations (e.g., MERRA-2 reanalysis data, AERONET ground-based measurements) could be incorporated to better constrain retrieval parameters in regions affected by aerosol-cloud interactions; c) Development of advanced retrieval algorithms: More sophisticated methods, such as machine learning or deep learning approaches, could be employed to better represent the complex relationships between land surface, atmosphere, and clouds.

Thank you once again for guiding the direction of our discussion, which has significantly enhanced the comprehensiveness of this study.

**Detailed modifications are as follows:** Based on your suggestions, we have incorporated relevant discussions in Section 5 (Discussion and Conclusion) of the manuscript. "Meanwhile, the validation results indicate that the RMSE of stratiform cloud profile structural characteristics over land is significantly higher than that over sea. This discrepancy is considered to be mainly attributable to the following factors: a) Sub-pixel Surface Heterogeneity: Variations in surface reflectance among different land cover types (e.g., vegetation, bare soil, urban areas) lead to mixed-pixel effects, complicating the decoupling of cloud optical properties. b) Aerosol Interference: Higher and spatiotemporally variable aerosol loadings over land can perturb cloud signals either indirectly by altering cloud microphysics (e.g., through cloud condensation nuclei effects) or directly via scattering. c) Surface Heating Effects: The lower thermal inertia of land surfaces results in more complex boundary-layer dynamics, increasing spatiotemporal variability in cloud base height and cloud layer thickness, which in turn elevates retrieval uncertainty. d) Interference from complex terrain and high-albedo surfaces: Complex terrain (e.g., mountains) and high-albedo surfaces (e.g., snow cover) are prone to causing false positives in cloud detection or overestimation of optical thickness. It is suggested that the following strategies could be adopted in the future to improve the estimation accuracy of stratiform cloud profile structural characteristics over land: a) Integration of land cover classification data (e.g., MODIS Land Cover product); b) Integration of aerosol ancillary data: Multi-source aerosol observations (e.g., MERRA-2 reanalysis data,

AERONET ground-based measurements) could be incorporated to better constrain retrieval parameters in regions affected by aerosol-cloud interactions; c) Development of advanced retrieval algorithms: More sophisticated methods, such as machine learning or deep learning approaches, could be employed to better represent the complex relationships between land surface, atmosphere, and clouds."

**Minor Comment 1:** Vertical resolution impact on thin layers:

While CloudSat's 240-m resolution is mentioned (Sec. 2.1), its inability to resolve sub-240 m layers (e.g., thin stratus) should be explicitly discussed.

**Response:** Thank you for your suggestions, which have made the discussion on payload limitations more comprehensive. Following your advice, we have supplemented Section 2.1 with a discussion about CloudSat's Cloud Profile Radar being unable to resolve cloud layers thinner than 240 m (thin clouds) due to its inherent resolution constraints. The modifications have been made as indicated above.

**Detailed modifications are as follows:** We supplemented and explained this limitation of CloudSat at the end of Section 2.1 (CloudSat data). "and (3) due to the limitations of 240-m resolution, it may not be possible to identify ultra-thin layer structures below 240m."

**Minor Comment 2:** Terminology consistency:

Line 218: "Stratiform water cloud profiles" → "liquid cloud profiles" for consistency with the rest of the paper.

**Response:** Thank you for your correction. We have revised the terminology on line 218 of the original text and conducted a full-text review to ensure terminological consistency throughout the manuscript.

**References:**

Buggee, A. J. and Pilewskie, P. A.: Retrieving Vertical Profiles of Cloud Droplet Effective Radius using Multispectral Measurements from MODIS: Examples and Limitations, EGUsphere, 2025, 1-27, 10.5194/egusphere-2025-546, 2025.

Du, J., Ge, J., Zhang, C., Su, J., Hu, X., Zhu, Z., Li, Q., Huang, J., and Fu, Q.: An Accurate Retrieval of Cloud Droplet Effective Radius for Single-Wavelength Cloud Radar, IEEE Transactions on Geoscience and Remote Sensing, 62, 1-11, 10.1109/TGRS.2024.3447002, 2024.

Ma, R. and Husi, L.: CARE product: Cloud microphysics and shortwave radiation forcing algorithms and applications., National Remote Sensing Bulletin, 28, 2320-2334, 10.11834/jrs.20232450, 2024.

Platnick, S.: Vertical photon transport in cloud remote sensing problems, Journal of Geophysical Research: Atmospheres, 105, 22919-22935, https://doi.org/10.1029/2000JD900333, 2000.

Shang, H., Chen, L., Bréon, F. M., Letu, H., Li, S., Wang, Z., and Su, L.: Impact of cloud horizontal inhomogeneity and directional sampling on the retrieval of cloud droplet size by the POLDER instrument,

Atmos. Meas. Tech., 8, 4931-4945, 10.5194/amt-8-4931-2015, 2015.

Shang, H., Letu, H., Wei, L., Ma, R., Wang, Y., Cai, Z., Yin, S., and Shi, C.: Remote sensing of liquid cloud profiles based on an analytical cloud profiling model, Science China Earth Sciences, 68, 998-1012, 10.1007/s11430-024-1532-2, 2025.

---

## Author Response (AR2)

Dear Prof. Zhang,

We sincerely thank you and the two referees for the time and effort spent reviewing our manuscript and providing constructive comments. Your feedback is invaluable for improving the quality of our paper. We fully understand and agree with your emphasis on the need for transparency, reproducibility, and uncertainty analysis regarding the cloud-base height (CBH) retrieval method used in this study. We have carefully addressed each of the four points you raised and made corresponding revisions to the manuscript, as detailed below.

**Editor Commet 1:** Add a Methods + Supplement section that documents the CBH algorithm in sufficient detail for reproduction (data and matchups; predictors/features; model architecture and hyperparameters; training/validation design and leakage controls; metrics by regime; and inference steps).

**Response:** We have incorporated a description of the CBH algorithm into Section 3 (Methodology) of the manuscript, as suggested. In addition, a detailed documentation of the algorithm has been provided in the Supplement in the form of a PDF file for reproduction. This document includes: Data and Matchups; Predictors/Features; Model Architecture and Hyperparameters; Training/Validation Design and Leakage Controls; Metrics by Regime; and Inference steps.

**Detailed modifications are as follows:** In the revised Section 3 (Methodology), we have added the following comprehensive description of the cloud base height (CBH) retrieval algorithm:

3.6 Cloud base height retrieval algorithm

The cloud base height (CBH) retrieval algorithm employed in this study is based on a deep neural network trained on Parasol L1 measurements collocated active sensor observations. The training dataset comprises Parasol L1 data—including intensity from 14 viewing angles in the oxygen A-band (763 nm and 765 nm channels), longitude, latitude, elevation, and cloud indicator—along with the corresponding CBH values and cloud detection information obtained from the CloudSat-CALIPSO L2 product 2B-CLDCLASS-LIDAR. Data from March, June, September, and December 2007 are primarily used, with the last seven days of each month reserved for testing and the remaining data used for training. To ensure high-quality training data, only cases where Parasol confidently detected cloudy scenes and CloudSat identified single-layer clouds are retained. Spatial collocation accuracy is constrained to within 0.01°, while temporal discrepancies are negligible due to the near-simultaneous observations from the A-Train satellites. The model utilizes geographic coordinates (longitude, latitude, elevation) and multi-angle oxygen A-band information from Parasol as inputs to predict CBH, with CloudSat-derived heights serving as ground truth. This method enables the CBH retrieval using only passive observations as input. The validation results indicate that the retrieval achieves a mean absolute error (MAE) of 0.78 km, a bias of 0.22 km, and a correlation coefficient (R) of 0.82.

It is important to note that the development, comprehensive validation, and detailed methodological discussion of this algorithm are beyond the scope of this study. A full description of the algorithm has been submitted to a separate journal and is currently under review. To ensure transparency and reproducibility, we provide a complete documentation of the algorithm in the Supplementary Material of this article. Furthermore, the code and pretrained model have been made publicly available (see the Data and Code Availability section). We have documented the CBH retrieval algorithm in detail in the Supplement, as follows:

**Supplement**

**S1. Methodology Details for Cloud Base Height Retrieval**

This method primarily retrieves cloud bottom height (CBH) by utilizing a deep neural network (DNN) based on multi-angle passive satellite observations—POLDER/Parasol data in the Oxygen A (O-A) band.

**1. Data and Matchups**

The data used in this study consist of four months (March, June, September, and December) of Parasol L1 products from 2007 and the corresponding CloudSat-CALIPSO 2B-CLDCLASS-LIDAR joint product. Spatial collocation between active and passive sensors maintains an accuracy within 0.01°, and temporal differences are negligible owing to the near-simultaneous measurements provided by the A-Train satellite constellation. During data preprocessing, only pixels that are identified as cloudy in Parasol data and simultaneously classified as single-layer clouds in the CloudSat-CALIPSO product are selected.

**2. Predictors/Features**

The training data used is shown in the table below.

**Table S1.** The training data and related parameters

| Satellite            | Product Name      | Parameters              | Note              |
|----------------------|-------------------|-------------------------|-------------------|
| Parasol              | L1                | longitude               |                   |
|                      |                   | latitude                |                   |
|                      |                   | elevation               |                   |
|                      |                   | cloud indicator         | Cloud detection   |
|                      |                   | I763NP (14 view angles) | OA band           |
|                      |                   | I765NP (14 view angles) | OA band           |
| CloudSat-
CALIPSO | 2B-CLDCLASS-LIDAR | Longitude               |                   |
|                      |                   | Latitude                |                   |
|                      |                   | CloudLayerBase          | Cloud base height |
|                      |                   | Cloudlayer              |                   |

**3. Model Architecture and Hyperparameters**

This method primarily employs deep neural networks to train the combined active and passive data, with specific parameter settings detailed in the table below.

**Table S2.** Parameter settings for training the model

| Parameter Category              | Parameter Name    | Value / Setting                  |
|---------------------------------|-------------------|----------------------------------|
| Randomness control              | random_state      | 42                               |
|                                 | dense_units       | [1024, 512, 256, 64]             |
| Network Architecture Parameters | dense_activations | ['relu', 'relu', 'relu', 'relu'] |
|                                 | output_activation | 'relu'                           |
| Training Parameters             | batch_size        | 21000                            |

| / Hyperparameters | epochs        | 44      |
|-------------------|---------------|---------|
|                   | optimizer     | 'adam'  |
|                   | loss_function | 'mae'   |
|                   | metrics       | ['mae'] |

**4. Training/Validation Design and Leakage Controls**

For each month, the last seven days of data serve as the test set, while the remaining days form the training set.

**5. Metrics by Regime**

The validation results of the model for predicting CBH across different cloud phases and various cloud types are as follows.

**Table S3.** The validation results for different cloud phases.

| Cloud phase | Error metric | Value   |  |
|-------------|--------------|---------|--|
|             | N            | 188282  |  |
|             | MAE          | 0.3831  |  |
| Water cloud | Bias         | -0.0532 |  |
|             | RMSE         | 0.9361  |  |
|             | R            | 0.6295  |  |
|             | N            | 76507   |  |
|             | MAE          | 0.6403  |  |
| Ice cloud   | Bias         | -0.1291 |  |
|             | RMSE         | 1.1839  |  |
|             | R            | 0.6877  |  |
|             | N            | 87373   |  |
|             | MAE          | 1.7632  |  |
| Mixed cloud | Bias         | 1.1279  |  |
|             | RMSE         | 3.1291  |  |
|             | R            | 0.7329  |  |

**Table S4.** The validation results for different cloud types.

| Cloud type  | Error metric | Value  | Cloud type    | Error metric | Value   |
|-------------|--------------|--------|---------------|--------------|---------|
|             | N            | 36577  |               | N            | 37464   |
|             | MAE          | 2.6885 |               | MAE          | 1.4915  |
| Cirrus      | Bias         | 2.1800 | Stratus       | Bias         | 0.5277  |
|             | RMSE         | 4.4079 |               | RMSE         | 2.0529  |
|             | R            | 0.1909 |               | R            | 0.6539  |
|             | N            | 27637  |               | N            | 9850    |
|             | MAE          | 1.1790 |               | MAE          | 0.1520  |
| Altostratus | Bias         | 0.2194 | Stratoculumus | Bias         | 0.0006  |
|             | RMSE         | 1.7428 |               | RMSE         | 0.2630  |
|             | R            | 0.6458 |               | R            | 0.8246  |
|             | N            | 153395 |               | N            | 47891   |
| Altoculumus | MAE          | 0.2653 | Culumus       | MAE          | 0.5238  |
|             | Bias         | 0.0051 |               | Bias         | -0.3217 |

| RMS | SE 0.4765 | RMSE | 1.4175 |
|-----|-----------|------|--------|
| R   | 0.7447    | R    | 0.2962 |

**6. Inference Steps**

The detailed steps for performing cloud base height prediction using the proposed method are as follows:

- Step 1. Read Parasol L1 data.
- Step 2. Apply quality control flags to filter cloudy pixels.
- Step 3. Extract all predictor variables specified in the "feature list" from the data.
- Step 4. Preprocess the extracted features exactly as done during the training stage.
- Step 5. Feed the preprocessed feature matrix into the loaded pre-trained DNN model.
- Step 6. Perform inference to obtain the estimated CBH for each pixel.

**Editor Commet 2:** Publicly archive the CBH materials (e.g., a preprint or methods note) and provide a stable citation. This is for transparency; the current paper must still be self-contained.

**Response:** The paper detailing the CBH retrieval algorithm has been submitted to *Remote Sensing of Environment* (Manuscript ID: RSE-D-25-02483) and is currently under review. Following your suggestion, we have made a preprint version of the manuscript available on Elsevier's Preprints (SSRN) platform to ensure a stable and citable reference. The full citation which has been added to reference of our manuscript is as follows:

Ji, T., Shang, H., Wei, L., Bao, F., Liu, Z., Wang, Y., Bao, S., Yin, S., Shi, C., Wang, H., Liu, Z., and Letu, H.: Retrieval of the base heights and cloud geometric thicknesses of clouds based on the PARASOL measurement, SSRN, https://ssrn.com/abstract=5515438 or http://dx.doi.org/10.2139/ssrn.5515438, 2025.

**Editor Commet 3:** Release code/model artifacts and a minimal dataset slice under a permanent DOI, and include a clear Data & Code Availability statement.

**Response:** We have archived the relevant code, pre-trained model, and a minimal input data slice for the CBH retrieval algorithm in a repository on *Zenodo*, which has been assigned a permanent DOI (10.5281/zenodo.17082185). A direct link to this repository has been added to the "Data and Code Availability" section of the manuscript to ensure full access for readers. Additionally, to protect the intellectual property associated with both this study and the CBH study prior to formal publication, the associated code and model resources are currently under restricted access (the right to view but no download) on *Zenodo*. The repository will be transitioned to fully public access immediately upon the formal acceptance of both manuscripts.

**Detailed modifications are as follows:** The modified "Data and Code Availability" section is as follows:

**Data and Code Availability**

All datasets used in this work are open-source. The CloudSat datasets are available from the

CloudSat Data Processing Center of the Cooperative Institute for Research in the Atmosphere (http://www.cloudsat.cira.colostate.edu/). The Parasol products are available from ICARE Data and Services Center (https://www.icare.univ-lille.fr/). The code and pre-trained model for the cloud base height retrieval algorithm, along with a minimal dataset required to reproduce the key results, have been deposited on Zenodo under a permanent DOI: [10.5281/zenodo.17082185].

Editor Commet 4: Provide an ablation/uncertainty analysis quantifying how CBH errors propagate into TP\_CER/TP\_NCOT and the reconstructed profiles (include a "perfect CBH" benchmark for context).

**Response:** Thank you very much for your valuable comment. The issue you raised regarding quantifying how CBH uncertainties propagate into the retrieval results is indeed crucial and represents a core aspect of our study. We fully agree with this point and have conducted a detailed ablation/uncertainty analysis accordingly. Below is an overview of our uncertainty analysis method and key findings:

This study quantifies how CBH errors propagate into TP\_CER/TP\_NCOT and the reconstructed profiles. We use a representative single-layer liquid stratiform cloud profile scene (from the track on March 2, 2007, corresponding to the profile data in Fig.9 of the original manuscript) as the test case. The CloudSat-measured CBH serves as the "perfect CBH" benchmark. We add Gaussian errors with standard deviations of 0.1 km and 0.5 km, along with systematic errors of 0.1 km and 0.5 km, to generate four perturbation experimental groups. These perturbed CBH values then serve as inputs for estimating TP\_CER and TP\_NCOT. The TP\_CER estimation results with added Gaussian errors and systematic errors are shown in Figures R1 and R2, respectively. Similarly, the TP\_NCOT estimation results with added Gaussian errors and systematic errors are presented in Figures R3 and R4, respectively. For clarity, we summarize all error evaluation metrics in Table R1.

For TP\_CER: As shown in Table 1 and Figures R1-R2, the addition of Gaussian errors of 0.1 km and 0.5 km to CBH increases the RMSE from 1.04 to 1.05 and 1.08, respectively, while R remains unchanged. When a systematic error of 0.1 km is added to CBH, the RMSE shows no change and remains at 1.04. With a systematic error of 0.5 km, the RMSE increases from 1.04 to 1.07. Throughout the addition of systematic errors, R continues to show no change. These results demonstrate that CBH errors within 0.5 km have negligible impact on TP\_CER estimation.

For TP\_NCOT: As shown in Table 1 and Figures R3-R4, the addition of a 0.5 km Gaussian error to CBH increases the RMSE from 0.08 to 0.09. In contrast, the addition of 0.1 km Gaussian error, 0.1 km systematic error, or 0.5 km systematic error produces no measurable change in RMSE. Furthermore, all variations in R remain within 0.01. We therefore conclude that CBH errors within 0.5 km have negligible impact on TP\_NCOT estimation.

In conclusion, these results demonstrate that CBH errors within 0.5 km have negligible impact on the estimation of both TP\_CER and TP\_NCOT. Consequently, the profile reconstruction is virtually unaffected by these errors. Based on the negligible impact observed in our sensitivity experiments, further analysis of CBH uncertainty effects on profile reconstruction is deemed unnecessary.

**Table R1.** The assessment results of the influence of cloud base height uncertainty on TP\_CER and TP\_NCOT

|         | Group       | Specific operation        | RMSE | R    | Bias  | MAE  |
|---------|-------------|---------------------------|------|------|-------|------|
|         | Control     | No error added to CBH     | 1.04 | 0.94 | 0.05  | 0.84 |
|         | group       |                           |      |      |       |      |
|         |             | CBH with gaussian error   | 1.05 | 0.94 | 0.05  | 0.84 |
|         | Gaussian    | (std=0.1km)               |      |      |       |      |
| TP_CER  | error group | CBH with gaussian error   | 1.08 | 0.94 | 0.03  | 0.87 |
|         |             | (std=0.1km)               |      |      |       |      |
|         |             | CBH with systematic error | 1.04 | 0.94 | -0.00 | 0.84 |
|         | Systematic  | (+0.1km)                  |      |      |       |      |
|         | error group | CBH with systematic error | 1.07 | 0.94 | -0.22 | 0.85 |
|         |             | (+0.5 km)                 |      |      |       |      |
|         | Control     | No error added to CBH     | 0.08 | 0.73 | -0.00 | 0.06 |
|         | group       |                           |      |      |       |      |
|         |             | CBH with gaussian error   | 0.08 | 0.75 | 0.00  | 0.05 |
|         | Gaussian    | (std=0.1km)               |      |      |       |      |
| TP NCOT | error group | CBH with gaussian error   | 0.09 | 0.64 | -0.01 | 0.06 |
| II_NCOI |             | (std=0.5km)               |      |      |       |      |
|         |             | CBH with systematic error | 0.08 | 0.76 | 0.01  | 0.05 |
|         | Systematic  | (+0.1 km)                 |      |      |       |      |
|         | error group | CBH with systematic error | 0.08 | 0.75 | 0.01  | 0.05 |
| -       |             | (+0.5km)                  |      |      |       |      |

**Figure R1.** Comparison of TP\_CER results between no error added to CBH and Gaussian error added to CBH. (a1) and (b1): no error added to CBH – "prefect" CBH; (a2) and (b2) CBH with gaussian error (std = 0.1km); (a3) and (b3) CBH with gaussian error (std = 0.5km).

**Figure R2.** Comparison of TP\_CER results between no error added to CBH and system error added to CBH. (a1) and (b1): no error added to CBH – "prefect" CBH; (a2) and (b2) CBH with system error (+0.1km); (a3) and (b3) CBH with system error (+0.5km).

**Figure R3.** Comparison of TP\_NCOT results between no error added to CBH and Gaussian error added to CBH. (a1) and (b1): no error added to CBH – "prefect" CBH; (a2) and (b2) CBH with gaussian error (std = 0.1km); (a3) and (b3) CBH with gaussian error (std = 0.5km).

**Figure R4.** Comparison of TP\_NCOT results between no error added to CBH and system error added to CBH. (a1) and (b1): no error added to CBH – "prefect" CBH; (a2) and (b2) CBH with system error (+0.1km); (a3) and (b3) CBH with system error (+0.5km).